# Source-Free Open-World RF Fingerprint Identification

**Kunling Li** [1]   **Cunqing Hua** [1]   **Hongyu Zhu** [1]   **Tianjie Ju** [1]   **Pengwenlong Gu** [2]

## Abstract

Radio Frequency Fingerprint Identification (RFFI) is a foundational pillar of physical-layer security, providing unclonable identity authentication and lightweight defense mechanisms for zero-trust wireless networks. Its practical deployment, however, often occurs in a source-free open-world (SF-OW) setting, characterized by a continuous influx of unregistered devices and privacy constraints that preclude the retention of historical data. In this paper, we formalize SF-OW RFFI task, which manifests a severe stability-plasticity dilemma: intrinsic signal similarity confuses new classes, while source absence precipitates catastrophic forgetting. To address this, we propose Incremental Orthogonal ETF (IO-ETF), a novel neural collapse-inspired framework utilizing output geometry to actively induce parameter separation and isolation. We further devise a Triple-Level Geometric Alignment (TLGA) strategy via semantic optimal transport, manifold progressive anchoring, and reliable subspace retention to stably align unlabeled streams to this geometric skeleton. Experiments on benchmarks demonstrate a superior trade-off between old-class retention and new-class discovery, offering a promising solution for secure access in dynamic networks.

## 1. Introduction

Next-generation wireless networks face significant challenges in establishing lightweight, intrinsic trust among heterogeneous devices (Cui et al., 2025). Traditional cryptographic authentication schemes incur high management costs and remain vulnerable to various attacks. Radio Frequency Fingerprint Identification (RFFI) has emerged as a

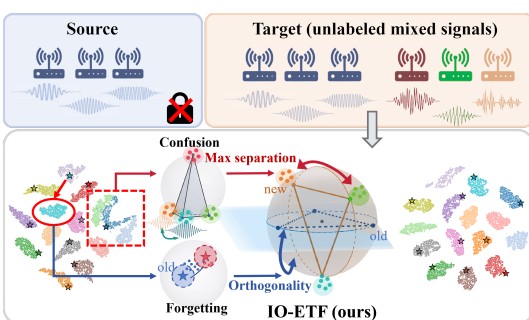

*Figure 1.* Top: SF-OW RFFI scenario with mixed unlabeled signals. Bottom: Visualizations of the stability-plasticity dilemma. Conventional adaptation suffers from new-class confusion (merged clusters) and old-class forgetting (manifold drift). Our geometric constraints achieve structural separation and task isolation.

promising non-cryptographic, physical-layer device authentication alternative (Zhang et al., 2025). It identifies devices by their subtle yet unique signal distortions (known as RF fingerprint (RFF)) caused by the hardware imperfections in transmitter (Cekic et al., 2021). This unclonable and lightweight paradigm is ideally suited for secure, efficient access control in the evolving wireless landscape.

Deep learning has been widely adopted to boost the RFFI accuracy (Li et al., 2025; Xie et al., 2025). However, two critical gaps hinder its real-world deployments. First, the prevailing closed-set RFFI solution is fragile against unregistered devices in open wireless environments. Moreover, to ensure security accountability, systems must go beyond simple outlier rejection (open-set) to granularly distinguishing among new unauthorized devices (open-world) for traceability. Second, and more critically, this expansion must be source-free: RFFs are permanent, biological-like identifiers, storing them directly creates traceable sensitive privacy and persistent threats. Therefore, RFFI can truly realize its physical-layer security advantages only if it can achieve open-world recognition without historical replay.

To bridge this gap, we formalize the Source-Free Open-World (SF-OW) RFFI problem, whereby a pre-trained model on old devices must adapt to an unlabeled stream containing both old and new transmitters. This exposes a severe stability-plasticity dilemma as shown in Figure 1: (i) *New-class confusion*: Without supervision, similarity-driven self-organization causes new features to collapse

[1]the School of Computer Science, Shanghai Jiao Tong University, Shanghai 200240, China [2]laboratory CEDRIC, Conservatoire national des arts et métiers, 75003 Paris, France. Correspondence to: Cunqing Hua <cqhua@sjtu.edu.cn>.

*Proceedings of the $43^{rd}$ International Conference on Machine Learning*, Seoul, South Korea. PMLR 306, 2026. Copyright 2026 by the author(s).

into adjacent or squeezed clusters, failing to form separable boundaries for new classes, even worse for fine-grained RFF discrepancies. (ii) *Catastrophic forgetting*: Adapting to the mixed stream drifts old-class representations and blurs established decision boundaries without replay. These stem from a fundamental conflict: the same gradient updates must concurrently separate new identities while preserving the feature manifolds of old classes. In the challenging setting of unlabeled mixed streams without replay, this conflict persistently triggers representation collapse and forgetting.

This paper resolves this dilemma by shifting the focus from parameter-space competition to output-space geometry. Motivated by the phenomenon of *neural collapse* (Papyan et al., 2020), we prove that fixing a strategically designed classifier head as decision anchors intrinsically ensures worst-case separability and steers gradient updates. To this end, we propose the Incremental Orthogonal Equiangular Tight Frame (IO-ETF) framework. This geometric skeleton enforces cross-block orthogonality to isolate gradient update interference between old and new classes, and minimizes intra-block coherence to create separable margins, thereby addressing both catastrophic forgetting and new-class confusion. To stably align unlabeled streams to this geometry, we propose Triple-Level Geometric Alignment (TLGA) mechanisms, which orchestrate semantic optimal transport to prevent assignment collapse via global balancing, progressive manifold anchoring to enable stable topology-preserving alignment, and reliability-driven pseudo-replay to pin old-class subspaces against drift.

In summary, our main contributions are as follows:

- We formalize the SF-OW RFFI problem and reframe its core stability-plasticity dilemma through the lens of output-space geometry, establishing that a fixed anchor geometry fundamentally governs both decision-boundary separability and gradient routing.

- We propose a two-part solution: (a) the IO-ETF framework, a neural collapse inspired geometry that isolates task interference to prevent forgetting and suppresses feature confusion; and (b) the TLGA mechanism, which achieves a tri-level alignment of unlabeled mixed streams to the optimal geometric skeleton.

- We evaluate the proposed scheme on multiple public benchmarks. The results demonstrate excellent identification performance and superior trade-off between old-class retention and new-class discovery.

## 2. Related Work

### 2.1. RFFI

RFFI has attracted growing attention for lightweight physical-layer device authentication (Li et al., 2025). DL-based RFFI presents notable advantages due to its strong end-to-end capability for extracting complex features with limited prior (Sun et al., 2025; Al-Shawabka et al., 2025; Zhao et al., 2024). Most RFFI studies, however, assume a closed-set paradigm. Open-set RFFI has consequently emerged to address dynamic environments prone to access by unknown transmitters and spoofing attacks. Prior work detects outliers via distance-thresholding (Wang et al., 2024), device-incremental meta-learning (Li et al., 2024), and model-and-data driven neural synchrony (Xie et al., 2021). However, these methods fail to distinguish fine-grained new identities. Some recent work attempts to extend RFFI to open-world settings (Han et al., 2025), but still relies on active supervision. In contrast, we address this deficiency with SF-OW RFFI to discover new classes from mixed unlabeled streams while preserving old-class recognition, relying solely on the pre-trained model.

### 2.2. Open-World Recognition

To model realistic distributions where known and novel classes co-exist (Guo et al., 2022), open-world recognition (OWR) has evolved from new class discovery (NCD) (Han et al., 2019) to generalized category discovery (GCD) (Vaze et al., 2022). Pioneering work ORCA (Cao et al., 2021) mitigates old-class bias via uncertainty gating, while later methods refine unknowns through pairwise similarity (Rizve et al., 2022) and contrastive consistency (Sun & Li, 2023). Recent work further addresses training bias via representation alignment (Xiao et al., 2024) and frequency decoupling (Wang et al., 2025). Despite these advances, these methods rely on similarity-driven self-clustering, which is prone to confusion and even assignment collapse. In contrast, we introduce neural collapse-inspired max-separation geometry as the optimal structural prior to shape representation space and mitigate confusion.

### 2.3. Source-Free Anti-Forgetting

Privacy and storage constraints often preclude access to source data, necessitating source-free learning. In Source-Free Domain Adaptation (SFDA), SHOT (Liang et al., 2020) freezes the classifier and self-trains the feature extractor. However, most SFDA assumes a fixed label space, lacking mechanisms for discovering multiple new classes. Moreover, updating representations to accommodate new classes without source replay risks catastrophic forgetting. Continual learning (CL) mitigates this via distillation (Li & Hoiem, 2017), parameter-importance constraints like EWC (Kirkpatrick et al., 2017) and MAS (Aljundi et al., 2018), and gradient projection like GPM (Saha et al., 2021), but these typically require new-class supervision or access to previous inputs, both absent in our unlabeled mixed setting. This paper shifts anti-forgetting paradigm from data-driven parameter regularization to structure-driven output-geometry

isolation, decoupling memory preservation from source data dependencies.

# 3. Theoretical Motivation

In source-free open-world recognition, the classifier head $\mathbf{M} = [m_1, \ldots, m_K]$ serves as the fundamental interface between feature manifolds and decision space. We refer to the angular structure in this last-layer space induced by these normalized anchors as the *Output Geometry*. In this section, we establish that this geometry governs learning dynamics in two directions: it determines the decision boundaries for inference and strictly dictates the gradient entry directions for back-propagation. This motivates two desiderata: constructing NC geometry to maximize separability for resisting confusion, and imposing subspace orthogonality to isolate gradients for mitigating interference-induced forgetting.

## 3.1. NC Geometry Maximizes Worst-case Separability

The recent work (Papyan et al., 2020; Zhu et al., 2021) has discovered an intriguing phenomenon called neural collapse (NC), which reveals that last-layer features and classifiers of a trained deep neural networks converge to a Simplex Equiangular Tight Frame (ETF), yielding intra-class collapse and maximal inter-class separation.

**Definition 3.1** (Simplex ETF). Given $K$ classes and feature dimension $d$, let the normalized classifier head $\mathbf{M} \in \mathbb{R}^{d \times K}$ with $\|m_k\|_2 = 1$. $\mathbf{M}$ is a Simplex ETF iff

$$\mathbf{M}^\top \mathbf{M} = \frac{K}{K-1}\left(\mathbf{I}_K - \frac{1}{K}\mathbf{1}_K\mathbf{1}_K^\top\right), \quad (1)$$

equivalently, $m_i^\top m_j = -\frac{1}{K-1}$ for all $i \neq j$.

We quantify worst-case separability via the signed coherence $\mu(\mathbf{M}) \triangleq \max_{i \neq j} m_i^\top m_j$. For $K \leq d+1$, the simplex bound gives $\min \mu(\mathbf{M}) \geq -\frac{1}{K-1}$, and the equality holds iff $\mathbf{M}$ is a simplex ETF. Hence it is minimax-optimal for worst-case pairwise separability.

Consider the normalized nearest-anchor classifier $\hat{y}(z) = \arg\max_k m_k^\top z$ with $\|z\|_2 = 1$. The robustness against confusion is bounded by the angular margin:

**Proposition 3.2** (Worst-case angular margin). *For ground truth $y$, the correct classification $\hat{y}(z) = y$ is guaranteed if the angular deviation satisfies $\angle(z, m_y) < \frac{1}{2}\arccos(\mu(\mathbf{M}))$ (a uniform worst-case sufficient condition over all $y$).* *(Proof in Appendix A.1)*

Therefore, fixing the output geometry to a Simplex ETF minimizes $\mu(\mathbf{M})$ and thus maximizes the decision safety radius, yielding an optimal geometric buffer against open-world confusion. This directly motivates using a pre-fixed ETF head while optimizing only the backbone; Yang et al.

(2022) also shows it can removes unnecessary head degrees of freedom, exhibiting strong discriminative behavior.

## 3.2. Orthogonal Geometry Induces Gradient Isolation

Note that fixing the output geometry controls not only the inference boundaries but also constrains the optimization pathway, as gradients can only enter the representation through the fixed head matrix $\mathbf{M}$. By locking this subspace for error back-propagation, the enforcement of inter-task anchor orthogonality compels all gradient updates into mutually orthogonal subspaces, thereby isolating task interference. This anti-forgetting strategy aligns with conventional gradient-projection based CL paradigms like GPM (Saha et al., 2021). However, it establishes an inverted control point: instead of estimating a null space from input-side covariance statistics, we directly construct an implicit null space through the prescribed output geometry, eliminating dependency on source data.

Formally, consider the back-propagation chain from loss $\mathcal{L}$ through logits $\ell = \mathbf{M}^\top z$ to the features $z$. Let $\delta \triangleq \nabla_\ell \mathcal{L}$ denote the logits-layer error vector. The feature gradient is

$$\nabla_z \mathcal{L} = \left(\frac{\partial \ell}{\partial z}\right)^\top \nabla_\ell \mathcal{L} = \mathbf{M}\delta \in \text{span}(\mathbf{M}). \quad (2)$$

This establishes that the gradient update is physically confined to the fixed anchor subspace.

By constructing new anchors to be orthogonal to the old ones ($\mathbf{M}_{\text{new}} \perp \mathbf{M}_{\text{old}}$), we obtain the following structural isolation mechanism:

**Proposition 3.3** (Geometric Gradient Isolation). *If the fixed output geometry satisfies inter-block orthogonality, i.e., $\mathbf{M}_{\text{old}}^\top \mathbf{M}_{\text{new}} = 0$, then the gradient component driven by new tasks ($\delta_{\text{new}}$) is automatically routed to the null space of the old task:*

$$\Pi_{\text{old}}(\nabla_z \mathcal{L}) = \Pi_{\text{old}}(\mathbf{M}_{\text{old}}\delta_{\text{old}}) + \underbrace{\Pi_{\text{old}}(\mathbf{M}_{\text{new}}\delta_{\text{new}})}_{=0}$$
$$= \mathbf{M}_{\text{old}}\delta_{\text{old}}. \quad (3)$$

*where $\Pi_{\text{old}}$ is the projection onto the anchor subspace $\mathcal{S}_{\text{old}} = span(\mathbf{M}_{\text{old}})$.* *(Proof in Appendix A.2)*

The geometric design mitigates interference at the source (logits $\rightarrow$ features interface). Updates within $\mathcal{S}_{\text{old}}$ are driven solely by the old-class probability mass $\delta_{\text{old}}$, while new-task errors are routed entirely to $\mathcal{S}_{\text{old}}^\perp$. Moreover, under the spectral alignment of NC, this source-level decomposition propagates to deeper layers (Proof in Appendix A.3).

Thus, an output geometry that maintains intra-task maximal separation and inter-task orthogonality can theoretically serve as a structural foundation to simultaneously resist OW confusion and mitigate interference-induced forgetting.

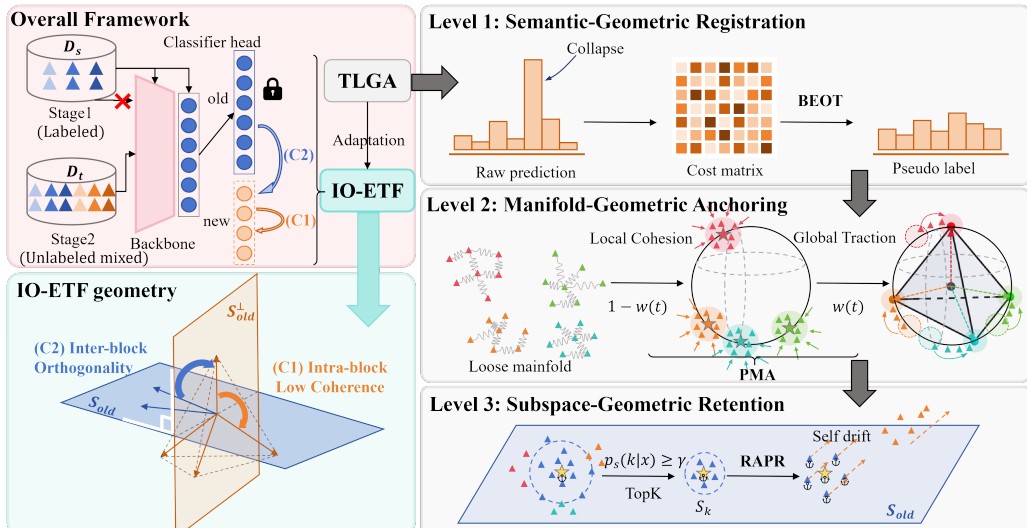

*Figure 2.* Overview of SF-OW RFFI framework. We resolve the stability-plasticity conflict via: (Left) IO-ETF expansion, which freezes the source head while inducing orthogonal (C2, isolation) and low-coherence (C1, max-separation) geometry; and (Right) TLGA, which stably maps unlabeled streams to this skeleton through a tri-level strategy: semantic (L1), manifold (L2), and subspace (L3).

## 4. Methodology

In this section, we formalize SF-OW RFFI problem, and present the proposed framework, with two core components: a tailored optimal geometric skeleton IO-ETF in Section 4.1, and a tri-level alignment mechanism TLGA in Section 4.2. The overall architecture is shown in Figure 2.

**Problem Definition.** Motivated by real-world RFFI deployments where privacy and storage constraints prevent source data retention despite the continuous arrival of new devices, we model the problem as a **Source-Free Open-World** setting with two stages. **Stage 1 (Source Pre-training):** A feature extractor $g_\theta$ and normalized classifier anchors $\mathbf{M}_{\text{old}} \in \mathbb{R}^{d \times K_{\text{old}}}$ are trained on a labeled source data set $\mathcal{D}_s = \{(x, y) \mid y \in \mathcal{Y}_{\text{old}}\}$, where $K_{\text{old}} = |\mathcal{Y}_{\text{old}}|$. **Stage 2 (Target Adaptation):** We access only an unlabeled target set $\mathcal{D}_t = \{x_i\}_{i=1}^N$ whose label space expands to $\mathcal{Y}_{\text{total}} = \mathcal{Y}_{\text{old}} \cup \mathcal{Y}_{\text{new}}$; throughout this stage, source data are inaccessible, and only the old model is available. Following prior settings (Cao et al., 2021; Fini et al., 2021), the number of new classes $K_{\text{new}} = |\mathcal{Y}_{\text{new}}|$ is assumed known. For any sample $x$ whose normalized representation is $z = g_\theta(x)/\|g_\theta(x)\|$, the unified classifier $\mathbf{M} = [\mathbf{M}_{\text{old}}, \mathbf{M}_{\text{new}}]$ predicts logits $\ell(x) = \mathbf{M}^\top z$. The goal is to retain performance on $\mathcal{Y}_{\text{old}}$ while discovering and separating $\mathcal{Y}_{\text{new}}$, yielding a unified decision rule over $\mathcal{Y}_{\text{total}}$.

### 4.1. Incremental Orthogonal ETF

**Active Geometric Prior.** While NC endows closed-set classification with an optimal ETF geometry (intra-class collapse and maximal inter-class margin), its natural emer-

gence is hindered in SF-OW target stage due to the lack of supervision and semantic drift. Therefore, we transition from the traditional view of NC as a passive training byproduct to deploying it as an active output-geometric prior: by explicitly fixing classifier anchors to a high-margin structure, we steer unstable representations towards a well-separated layout to structurally suppress open-world confusion.

However, directly employing a global Simplex ETF is incompatible with SF-OW. The simplex vertices depend on the total class number $K$; adding classes necessitates global polytope reconfiguration rather than local rotation. In the absence of $\mathcal{D}_s$, $g_\theta$ cannot be recalibrated to match this angular drift, which breaks the feature-anchor alignment for old classes established in Stage 1 and causes catastrophic forgetting. Moreover, analytical Simplex ETFs require $d \geq K-1$, which is not scalable to continuous growth of new classes.

**IO-ETF.** This framework is an anchor extension mechanism that requires no source data and avoids global reconstruction. We leverage theoretical insights in Section 3 as key knobs to resolve the SF-OW stability-plasticity dilemma by engineering the output geometry. During the target stage, we freeze the old anchors $\mathbf{M}_{\text{old}}$ and construct $\mathbf{M}_{\text{new}}$ for the unified head $\mathbf{M} = [\mathbf{M}_{\text{old}}, \mathbf{M}_{\text{new}}]$ to satisfy the following structural conditions:

**(C1) Intra-block Low Coherence (Anti-Confusion):** Minimize the coherence within $\mathbf{M}_{\text{new}}$ to maximize worst-case margins for new classes, thereby suppressing confusion, according to Prop. 3.2.

**(C2) Inter-block Orthogonality (Anti-Forgetting):** Enforce $\mathbf{M}_{\text{old}}^\top \mathbf{M}_{\text{new}} \approx \mathbf{0}$. This routes new-class errors to the

null space of the old anchors at the logits→features interface, according to Prop. 3.3, creating a "gradient firewall".

To satisfy (C1)-(C2) while bypassing dimension saturation limits ($d < K - 1$) and limited orthogonal-complement capacity ($d - \text{rank}(\mathbf{M}_{\text{old}})$), we propose an optimization-based IO-ETF construction inspired by Generalized Neural Collapse (GNC) (Liu et al., 2023b). We minimize a repulsive potential energy defined by maximum inner product on the unit hypersphere:

$$\min_{\mathbf{M}_{\text{new}}} \mathcal{E} = \underbrace{\sum_{\substack{i \in \mathcal{Y}_{\text{new}} \\ j \in \mathcal{Y}_{\text{old}}}} \phi(|m_i^\top m_j|)}_{\mathcal{E}_{\text{inter}}: \text{Stability (C2)}} + \lambda \underbrace{\sum_{\substack{k,l \in \mathcal{Y}_{\text{new}} \\ k \neq l}} \phi(m_k^\top m_l)}_{\mathcal{E}_{\text{intra}}: \text{Plasticity (C1)}}, \quad (4)$$

where $\|m_i\|_2 = 1, \forall i \in \mathcal{Y}_{\text{new}}$. We set $\phi(a) \triangleq \exp(\alpha a)$ with $\alpha > 0$, and for any collection $\{a_p\}$, $\frac{1}{\alpha} \log \sum_p \exp(\alpha a_p) \approx \max_p a_p$ for sufficiently large $\alpha$. Thus, Eq.(4) can be viewed as minimizing smooth upper bounds of both intra-block and inter-block coherence.

The inter-block $\mathcal{E}_{\text{inter}}$ penalizes non-orthogonality, driving $\mathbf{M}_{\text{new}}$ into the orthogonal complement of $\mathbf{M}_{\text{old}}$ for structural anti-forgetting (C2). $\mathcal{E}_{\text{intra}}$ drives new anchors to disperse uniformly, maximizing the worst-case margin against open-set confusion. In sufficient dimensions (where $\mathcal{S}_{\text{old}}^\perp$ can accommodate a $K_{\text{new}} - 1$ simplex), the solution recovers a Simplex ETF within the orthogonal complement. In oversaturated regimes ($K > d$), the optimization spontaneously yields a GNC low-coherence configuration, maintaining the theoretically maximum geometric margin under fixed dimension budgets, discussed in Section 5.3.

## 4.2. Tri-Level Geometric Adaptation

Given the optimal IO-ETF geometric skeleton, SF-OW reduces to a sample-to-slot assignment task: mapping unlabeled samples to pre-fixed geometric anchors (slots). However, driving entangled target mixture manifolds towards this rigid geometry introduces instability across three levels: semantics, manifold structure, and memory. To overcome these issues, we propose TLGA, a tri-level geometric adaptation framework.

**Level-1: Semantic-Geometric Registration.** Although IO-ETF defines the desired slot geometry, directly pseudo-labeling by per-sample $\arg\max p(x)$ without supervision leads to a winner-takes-all assignment collapse (Asano et al., 2019), where samples gravitate towards a few "low-resistance" slots, forming degenerate attractors and hindering NCD. Therefore, we replace local greedy assignment with batch-wise Balanced Entropic Optimal Transport (BEOT), reframing pseudo-labeling from a confidence competition to a capacity-constrained global matching.

Given a batch $\mathcal{B}$, define $\ell_i \triangleq \mathbf{M}^\top z_i$ and the batch logit

matrix $\mathbf{P} \triangleq [\ell_1, \ldots, \ell_b] \in \mathbb{R}^{K \times b}$, where $K = K_{\text{old}} + K_{\text{new}}$. We seek a soft assignment $\mathbf{Q} \in [0, 1]^{K \times b}$ by solving an entropic OT problem under capacity constraints:

$$\mathbf{Q}^* = \arg\max_{\mathbf{Q} \in \Gamma} \text{Tr}(\mathbf{Q}^\top \mathbf{P}) + \varepsilon \, \text{H}(\mathbf{Q}),$$

$$\text{s.t. } \Gamma = \left\{ \mathbf{Q} \in \mathbb{R}_+ \,\middle|\, \mathbf{Q}\mathbf{1}_b = \tfrac{1}{K}\mathbf{1}_K, \; \mathbf{Q}^\top\mathbf{1}_K = \tfrac{1}{b}\mathbf{1}_b \right\} \quad (5)$$

where $\text{H}(\cdot)$ denotes entropy and $\varepsilon > 0$ controls smoothness. $\Gamma$ enforces equiprobable activation over geometric slots within the batch (Caron et al., 2020; Fini et al., 2021), preventing representation collapse and ensuring cold-start gradients for immature clusters. $\mathbf{Q}^*$ can be efficiently computed via Sinkhorn-Knopp iterations (Cuturi, 2013), providing globally rectified pseudo-labels for subsequent geometric alignment.

**Level-2: Manifold-Geometric Anchoring.** After establishing semantic indexing, the target representation manifolds remain loose and topologically fragile. Forcing such coarse manifolds onto rigid IO-ETF anchors risks topological tearing, destroying local neighborhood structures crucial for NCD. To address this, we propose a Progressive Manifold Anchoring (PMA) scheme to dynamically balance local topology preservation and global geometric traction for stable, controllable manifold reshaping.

*Local Cohesion.* We maintain a streaming target feature bank $\mathcal{M}$. For each sample $z_i$ in the batch, we retrieve its $k$ nearest neighbor set $\mathcal{N}_i$ as soft positives from $\mathcal{M}$ based on cosine similarity $z_i^\top z$. Then we apply instance-level cohesive contrastive loss:

$$\mathcal{L}_{\text{local}} = -\frac{1}{|\mathcal{B}|} \sum_{i \in \mathcal{B}} \frac{1}{|\mathcal{N}_i|} \sum_{z^+ \in \mathcal{N}_i} \log \frac{\exp(z_i^\top z^+/\tau)}{\sum_{z \in \mathcal{M}} \exp(z_i^\top z/\tau)}. \quad (6)$$

This encourages local neighborhoods to cluster in topological space, preventing neighbor drift or topology tearing during global alignment.

*Global Traction.* As local topology solidifies, we leverage the OT-derived assignment $\hat{y}_i = \arg\max_k \mathbf{Q}^*_{k,i}$ to pull features towards IO-ETF anchors $m_{\hat{y}_i}$ via temperature-scaled cross-entropy:

$$\mathcal{L}_{\text{global}} = -\frac{1}{|\mathcal{B}|} \sum_{i \in \mathcal{B}} \log \frac{\exp(z_i^\top m_{\hat{y}_i}/\tau)}{\sum_{k=1}^{K} \exp(z_i^\top m_k/\tau)}. \quad (7)$$

This aligns clusters to the target IO-ETF slots while repelling others, actively inducing NC. The old slots alignment preserves old-class discriminability, while stable occupancy and aggregation on new slots supports NCD.

To address the trade-off between topology preservation and geometric slot alignment, we introduce a soft-to-hard evolutionary objective function:

$$\mathcal{L}_{\text{align}}(t) = w(t)\,\mathcal{L}_{\text{global}} + \big(1 - w(t)\big)\,\mathcal{L}_{\text{local}}. \quad (8)$$

*Table 1.* SF-OW RFFI performance comparison with other methods across datasets. (**Bold**: Best; Underline: Second-best).

| Type | Method | LoRa | | | | | Oracle | | | | | WiSig | | | | | *Overall Avg.* | | |
|---|---|---|---|---|---|---|---|---|---|---|---|---|---|---|---|---|---|---|---|
| | | Old | New | All | H-sc. | AUROC | Old | New | All | H-sc. | AUROC | Old | New | All | H-sc. | AUROC | Old | New | H-sc. |
| **Ref** | Teacher | 96.14 | 66.59 | - | 78.68 | - | 99.97 | 60.62 | - | 75.47 | - | 99.97 | 69.90 | - | 82.28 | - | 98.69 | 65.70 | 78.81 |
| **CL** | ST Only | 95.12 | 84.61 | 93.01 | 89.56 | 67.93 | 71.83 | 99.48 | 96.51 | 83.42 | 67.69 | 99.72 | 95.18 | 98.23 | 97.40 | 99.92 | 88.89 | 93.09 | 90.13 |
| | ST + LwF | 96.89 | 85.68 | 94.32 | 90.94 | 82.52 | 72.33 | 98.04 | 96.43 | 83.25 | 68.97 | 99.72 | 96.64 | 98.71 | 98.16 | 99.97 | 89.65 | 93.45 | 90.78 |
| | ST + EWC | 95.06 | 84.61 | 92.97 | 89.53 | 75.25 | 72.19 | 97.56 | 96.19 | 82.98 | 60.03 | 99.75 | 96.58 | 98.71 | 98.14 | 99.90 | 89.00 | 92.92 | 90.22 |
| | ST + MAS | 95.04 | 84.84 | 93.00 | 89.65 | 76.80 | 72.91 | 98.92 | 96.19 | 83.95 | 52.88 | 99.72 | 96.83 | 98.77 | 98.25 | 99.96 | 89.22 | 93.53 | 90.62 |
| | ST + GPM | 95.06 | 86.25 | 92.79 | 90.44 | 67.45 | 72.08 | 97.42 | 96.19 | 82.86 | 58.27 | 99.91 | 97.08 | 98.98 | 98.47 | 99.95 | 89.02 | 93.58 | 90.59 |
| | *Avg. (CL)* | 95.43 | 85.20 | 93.22 | 90.02 | 73.99 | 72.27 | 98.28 | 96.30 | 83.29 | 61.57 | 99.76 | 96.46 | 98.68 | 98.08 | 99.94 | 89.16 | 93.31 | 90.47 |
| **OW** | SHOT | 95.46 | 74.00 | 91.17 | 83.37 | 57.65 | 99.97 | 64.92 | 88.30 | 78.72 | 55.83 | 99.94 | 97.91 | 99.27 | 98.91 | 67.79 | 98.46 | 78.94 | 87.00 |
| | SF-GCD | 95.44 | 79.09 | 92.17 | 86.50 | 81.28 | 99.70 | 72.78 | 90.73 | 84.14 | 64.84 | 99.66 | 95.12 | 98.17 | 97.34 | 99.86 | 98.27 | 82.33 | 89.33 |
| | SF-OpenLDN | 97.37 | 67.89 | 91.47 | 80.00 | **97.00** | 99.97 | 82.30 | 94.08 | 90.28 | 90.57 | 99.38 | 76.43 | 91.83 | 86.41 | 83.13 | 98.91 | 75.54 | 85.56 |
| | SF-ORCA | 94.07 | **87.75** | 92.81 | 90.80 | 62.21 | 99.97 | 61.22 | 87.05 | 75.94 | 59.14 | 99.97 | 97.34 | 99.10 | 98.64 | 56.90 | 98.00 | 82.10 | 88.46 |
| | SF-OpenNCD | 95.14 | 84.48 | 93.01 | 89.49 | 71.35 | 72.08 | 99.68 | 96.43 | 83.66 | 69.12 | 99.75 | 97.72 | 99.08 | 98.72 | **99.98** | 88.99 | 93.96 | 90.62 |
| | *Avg. (OW)* | 95.50 | 78.64 | 92.13 | 86.03 | 73.90 | 94.34 | 76.18 | 91.32 | 82.55 | 67.90 | 99.74 | 92.90 | 97.49 | 96.00 | 81.53 | 96.53 | 82.57 | 88.19 |
| **-** | **Ours** | **97.62** | 85.85 | **94.64** | **91.36** | 93.38 | **100.00** | **99.69** | **99.92** | **99.84** | **99.97** | **99.98** | **99.90** | **99.96** | **99.94** | 99.73 | **99.20** | **95.15** | **97.05** |
| **Improv.** | vs. Best | +0.25 | -1.90 | +0.32 | **+0.42** | -3.62 | +0.03 | +0.01 | +3.41 | **+9.56** | +9.40 | +0.04 | +1.99 | +0.69 | **+1.03** | -0.24 | +0.29 | +1.19 | **+6.27** |
| | vs. Mean | +2.16 | +3.93 | +1.96 | **+3.34** | +19.43 | +16.69 | +12.46 | +6.11 | **+16.92** | +35.23 | +0.23 | +5.22 | +1.87 | **+2.90** | +9.00 | +6.35 | +7.21 | **+7.72** |

The weighting factor $w(t)$ governs a phased transition from local to global training. Initially, $\mathcal{L}_{\mathrm{local}}$ drives stable cluster formation with $w(t)$ close to 0. As $w(t) \to 1$, $\mathcal{L}_{\mathrm{global}}$ dominates, pulling these clusters toward the IO-ETF slots for a smooth evolution toward optimal geometry.

**Level-3: Subspace-Geometric Retention.** By Prop. 3.3, gradients on $\mathcal{S}_{\mathrm{old}}$ contain both cross-subspace interference and within-subspace drift. IO-ETF orthogonality largely isolates old-new interference, yet pseudo-label noise in mixed unlabeled streams inevitably triggers implicit self-drift ($\delta_{\mathrm{old}} \neq 0$), pushing old features off $\mathcal{S}_{\mathrm{old}}$ and causing forgetting. To counter this without source data, we propose Reliability-Aware Pseudo-Replay (RAPR), which uses the frozen source model as teacher to mine high-purity old-class cores and anchor them to old slots.

Let $p_{\mathrm{s}}(k \mid x)$ denote the teacher posterior for $k \in \mathcal{Y}_{\mathrm{old}}$. For each old class $k$, we apply dual screening (confidence thresholding and top-$k$ truncation) to select reliable samples from the candidate set $\mathcal{B}_k = \{x_i \in \mathcal{B} : \arg\max_{c \in \mathcal{Y}_{\mathrm{old}}} p_{\mathrm{s}}(c|x_i) = k\}$:

$$\mathcal{S}_k = \mathrm{TopK}_{\kappa}\left(\{x \in \mathcal{B}_k \mid p_{\mathrm{s}}(k|x) \geq \gamma\}, \; p_{\mathrm{s}}(k|x)\right), \quad (9)$$

where $\gamma$ filters low-confidence noise and $\mathrm{TopK}_{\kappa}(\mathcal{X}, s)$ returns the top $\kappa$ fraction of $\mathcal{X}$ ranked by score $s$, adaptively focusing on the high-density regions of old classes.

We then impose an old-class retention loss on the pseudo-replay set $\mathcal{A} = \bigcup_{k \in \mathcal{Y}_{\mathrm{old}}} \{(x, k) : x \in \mathcal{S}_k\}$:

$$\mathcal{L}_{\mathrm{ret}} = -\frac{1}{|\mathcal{A}|} \sum_{(x,k) \in \mathcal{A}} \log p(k|x). \quad (10)$$

This complements IO-ETF: orthogonal geometry handles cross-subspace interference, while $\mathcal{L}_{\mathrm{ret}}$ restricts subspace self-drift, jointly ensuring source-free old-class retention.

**Overall Objective.** The final objective is $\mathcal{L}_{\mathrm{total}}(t) = \mathcal{L}_{\mathrm{align}}(t) + \lambda_{\mathrm{old}}\mathcal{L}_{\mathrm{ret}}$. Through this tri-level adaptation, we achieve slot assignment and progressively align target manifolds to optimal IO-ETF geometry while preserving local topology and old-class retention, stably achieving old-class recognition and NCD in SF-OW scenarios.

## 5. Experiments

### 5.1. Experimental Setup

**Benchmarks and Metrics.** We evaluate our SF-OW RFFI methods on three representative RFF benchmarks with distinct protocols: WiSig (Hanna et al., 2022), Oracle (Sankhe et al., 2019), and LoRa (Shen et al., 2022), with classes partitioned into old/new sets. Raw signals are converted to channel-invariant spectrograms as network inputs. To evaluate performance of SF-OW RFFI, we utilize metrics including Old accuracy, New and All accuracies (via Hungarian matching), H-score for stability-plasticity trade-off, and AUROC for open-set separability. Detailed dataset statistics with preprocessing configurations and evaluation metrics can be found in the Appendix B.1 and B.2, respectively. Our code is available at `https://github.com/likunling-jingjing/SFOW_RFFI`.

**Implementation and Baselines.** We employ a uniform ResNet-18 backbone ($d = 512$) and compare against the Stage 1 pre-trained model (Teacher) and representative CL and OW methods adapted for SF-OW regime consis-

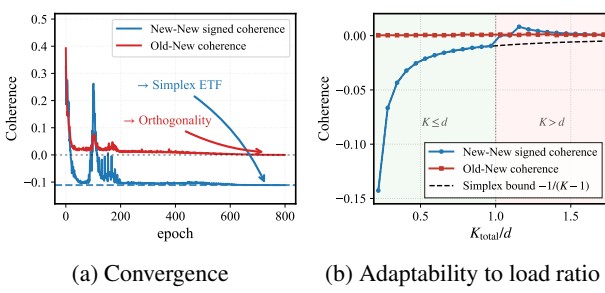

(a) Convergence       (b) Adaptability to load ratio

*Figure 3.* IO-ETF coherence convergence and variation across different dimensional load ratios $K_{total}/d$.

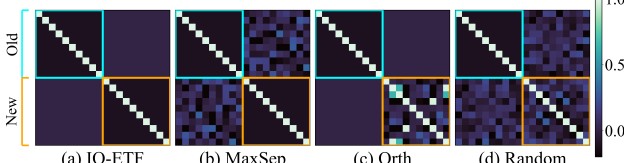

(a) IO-ETF    (b) MaxSep    (c) Orth    (d) Random

*Figure 4.* Visualization of anchor correlation matrices under different geometric constraints.

*Table 2.* Ablation on geometric constraints.

| Method | Oracle | | | | | WiSig | | | | |
|---|---|---|---|---|---|---|---|---|---|---|
| | Old | New | All | AUROC | Leak | Old | New | All | AUROC | Leak |
| Global | 0.02 | 62.92 | 53.07 | 61.50 | 0.23 | 0.07 | 80.72 | 74.38 | 66.22 | 0.20 |
| MaxSep | 93.67 | 89.24 | 88.34 | 90.52 | 0.55 | 88.42 | 99.11 | 91.94 | 89.76 | 0.45 |
| Random | 92.31 | 87.02 | 85.77 | 75.22 | 0.49 | 90.60 | 86.98 | 86.54 | 74.06 | 0.42 |
| Orth | **98.45** | 80.34 | 86.19 | 89.53 | **0.04** | **99.75** | 84.09 | 88.79 | **99.59** | **0.12** |
| **IO-ETF** | 96.25 | **97.30** | **92.43** | **96.60** | **0.04** | 98.20 | **99.75** | **98.71** | 98.76 | 0.17 |

tently: (i) CL methods, including LwF (Li & Hoiem, 2017), EWC (Kirkpatrick et al., 2017), MAS (Aljundi et al., 2018), and GPM (Saha et al., 2021), incorporate self-training for new discovery; (ii) OW methods, including GCD (Vaze et al., 2022), OpenLDN (Rizve et al., 2022), ORCA (Cao et al., 2021), OpenNCD (Liu et al., 2023a), and DA-based SHOT (Liang et al., 2020) are made source-free via teacher-guided reliability-filtered pseudo-labeling. Detailed baseline configurations, implementation and hyperparameter settings are provided in Appendix B.3 and B.4, respectively.

## 5.2. Comparison with Other Methods

We first evaluate SF-OW performance on RFFI benchmarks comparing with SOTA CL and OW baselines. As shown in Table 1, our method achieves SOTA, with 99.20% known and 95.15% novel accuracy, yielding the highest H-score of 97.05% (+6.27% over the best baseline). Notably, on Oracle, our scheme outperforms the runner-up (SF-OpenLDN) by +9.56%, confirming that our approach achieves the optimal stability-plasticity trade-off for SF-OW RFFI.

It can be observed that most baselines optimize either old or new performance but not both. For instance, on LoRa, SF-ORCA slightly improves plasticity (+1.9%) at the expense of stability (-3.55%). Specifically, CL methods (e.g., GPM) rely on accurate old-class inputs; in SF regimes, they fail to protect old boundaries under noisy self-training, with Oracle dropping to around 72%, causing catastrophic forgetting. Moreover, OW methods further degrade without supervision, struggling to separate similar features and cause new-class confusion (the average new accuracy drops to 82.57%) and boundaries erode with AUROC deteriorating in SHOT, where new samples are absorbed into old-class regions. In contrast, our method effectively resolves this dilemma, achieving the best overall trade-off.

**Failure Cases.** We further inspect the remaining errors via per-class diagnosis (detailed confusion matrix analysis in Appendix C.1). The failures are localized rather than global: most errors arise from a few physically similar borderline devices, such as asymmetric old-new confusion (N23→O6)

and hard new-new confusion (N21↔N22), rather than catastrophic forgetting or representation collapse.

## 5.3. Geometric Analysis of IO-ETF

We investigate the geometry induced by the optimization objective in Eq. (4). As shown in Figure 3, IO-ETF exhibits spontaneous structural emergence and adaptive optimality. In Figure 3(a), the new-new signed coherence rapidly converges to the theoretical Simplex lower bound $-1/(K-1)$, indicating that new anchors spontaneously organize into a maximally separated symmetric configuration within their effective subspace. Meanwhile, the old-new coherence vanishes to zero, stably decoupling the new subspace from old anchors. This confirms that the model automatically discovers the optimal IO-ETF geometry tailored for SF-OW: block-wise orthogonality for interference isolation and within-block maximal separation for discriminability.

We further investigate asymptotic behaviors of IO-ETF across varying load ratios in Figure 3(b). In the unsaturated regime ($K_{\text{total}} \leq d$), IO-ETF stably adheres to the *Block-orthogonal + Simplex-ETF* geometry. Once entering dimensional saturation ($K_{\text{total}} > d$) where the simplex bound is theoretically unattainable, the absolute coherence $\mu_{\text{abs}}(\mathbf{M}_{\text{new}}) = \max_{i \neq j} |\langle m_i, m_j \rangle|$ jumps to 1. This signals the emergence of antipodal pairs, marking the shift of new-block geometry from Simplex-ETF towards Orthoplex, consistent with the structure switch observed in GNC. This transition allows for denser spherical packing under dimensional constraints, while maintaining cross-block isolation, validating that IO-ETF can spontaneously select the globally optimal separation configuration.

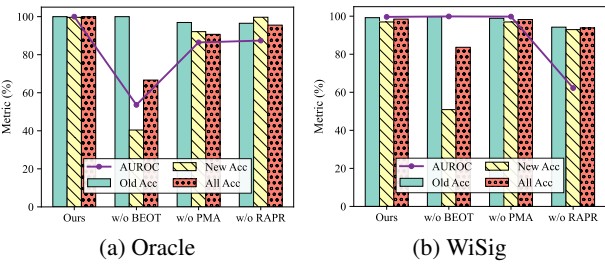

| (a) Oracle | (b) WiSig |
|---|---|

*Figure 5.* Ablation on three-level geometric alignment.

| (a) Oracle | (b) WiSig |
|---|---|

*Figure 6.* Model scalability to varying new class ratios.

## 5.4. Ablation on Geometric Constraints

We ablate geometric constraints to verify that output geometry governs gradient routing and decision-boundary formation, thereby affecting SF-OW performance. We compare IO-ETF against Global-ETF reconstruction, Random, MaxSep (Intra-ETF only), and Orth (Inter-Orthogonal only). Table 2 reports identification accuracies and the interference metric $\text{Leak} = \mathbb{E}\left[\frac{\|\Pi_{\text{old}}(\nabla_z L_{\text{new}})\|_2}{\|\nabla_z L_{\text{new}}\|_2}\right]$. Results confirm that geometric constraints explicitly control two distinct behaviors. First, inter-orthogonality is critical for gradient decoupling. Orth minimizes leakage (0.04) and preserves old accuracy (98.45%) via geometry isolation. However, without separation guidance, its new accuracy is limited (80.34%). Second, intra-separation drives plasticity. MaxSep boosts new accuracy (89.24%) by guiding separation, but suffers severe leakage (0.55) without an orthogonal firewall. Our IO-ETF attains the best overall performance, indicating that both constraints are necessary.

To uncover structural origins of these results, we visualize the anchor correlation matrices in Figure 4. Orth establishes a clear firewall but exhibits collapse of new anchors into highly similar directions, explaining its poor new discriminability. Conversely, MaxSep forms distinct anchors but noisy off-diagonal correlations, verifying the source of high leakage. IO-ETF uniquely achieves both a clean orthogonal barrier and strong intra-class separation, effectively translating structural advantages into superior SF-OW performance.

## 5.5. Ablation on TLGA

To validate the necessity of TLGA and its efficacy in suppressing specific structural failure modes, we conduct ablation by sequentially removing BEOT, PMA, and RAPR. As shown in Figure 5, the full TLGA consistently outperforms all variants. Removing semantic-level BEOT triggers severe assignment collapse. On Oracle, new accuracy plummets to 40.41%, indicating that global capacity constraints by OT are critical to prevent degenerate solutions and activate effective registration. Moreover, removing PMA results in topological tearing. Especially for the loosely-structured Oracle, forcibly pulling features toward rigid anchors destroys local neighborhood structures. This exacerbates inter-class confusion, dropping AUROC by around 14%. Finally, without RAPR subspace retention, although IO-ETF already blocks cross-task interference, the unsupervised mixed stream still induces old-class self-drift, yielding an around 5% accuracy drop. This demonstrates that anti-forgetting relies on both external isolation and internal anchoring. Overall, these three components synergistically ensure alignment across semantic boundaries, topological continuity, and subspace stability, forming a robust closed-loop framework. Results across datasets are provided in Appendix C.2.

## 5.6. Scalability Analysis

In Figure 6, we evaluate model robustness under varying new class ratios and compare H-score and AUROC against the best baseline (SF-OpenLDN). While performance declines as openness increases, our method consistently outperforms the baseline across all ratios. Notably, even under high new-class regime, our method maintains robust performance with H-score and AUROC over 80%, whereas baselines suffer severe degradation below 40%, validating the scalability benefits of our geometric adaptation. Results across datasets are detailed in Appendix C.3.

## 6. Conclusion

In this paper, we propose a novel SF-OW RFFI framework, offering a new perspective that shifts from parameter-space competition to proactive output-geometric design. By theoretically revealing that output geometry jointly governs worst-case separability and gradient routing, we repurpose NC phenomenon as an active prior to construct IO-ETF. This superior geometric skeleton structurally resolves the dual SF-OW failure modes of confusion and forgetting by maximizing decision boundaries and enforcing subspace isolation. We further propose a triple-level alignment strategy to stably align unlabeled streams to this rigid geometry. Experiments validate the superior effectiveness and robustness of our approach, establishing an optimal stability-plasticity trade-off. We hope this work can inspire further exploration of explicit geometric priors in SF-OW learning.

## Acknowledgements

This work was supported by the Natural Science Foundation of China under Grant U2572202 and the ANR V2XAntiJam (ANR-25-CE25-6505) project.

## Impact Statement

This paper presents work whose goal is to advance the field of Machine Learning. There are many potential societal consequences of our work, none which we feel must be specifically highlighted here.

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

# Appendix

In appendix, we provide detailed theoretical proofs, comprehensive implementation details, and additional experimental results and corresponding analyses to support the main paper. The content is organized as follows:

## A. Proofs

### A.1. Proof of Proposition 3.2

**Proposition 3.2 (Worst-case Angular Margin).** For ground truth $y$, the correct classification $\hat{y}(z) = y$ is guaranteed if the angular deviation satisfies $\angle(z, m_y) < \frac{1}{2} \arccos\bigl(\mu(\mathbf{M})\bigr)$ (a uniform worst-case sufficient condition over all $y$).

*Proof.* By construction, all anchors $m_i$ and features $z$ are $\ell_2$-normalized. On the unit hypersphere $\mathbb{S}^{d-1}$, define the angular (geodesic) distance

$$d(a, b) \triangleq \angle(a, b) = \arccos(a^\top b), \qquad a, b \in \mathbb{S}^{d-1}. \tag{11}$$

This distance equals the length of the shortest great-circle arc between two points on $\mathbb{S}^{d-1}$, hence $d$ is a metric on $\mathbb{S}^{d-1}$ and satisfies the triangle inequality. Moreover, for any unit vectors $u, v$, we have $u^\top v = \cos(d(u, v))$, and because $\cos(\cdot)$ is strictly decreasing on $[0, \pi]$, the normalized nearest-anchor rule

$$\hat{y}(z) = \arg\max_k m_k^\top z \tag{12}$$

is equivalent to selecting the closest anchor under $d$, i.e., $\hat{y}(z) = \arg\min_k d(m_k, z)$.

Fix any ground-truth class $y$ and any $j \neq y$. By the definition of signed coherence $\mu(\mathbf{M}) \triangleq \max_{i \neq j} m_i^\top m_j$, we have $m_y^\top m_j \leq \mu(\mathbf{M})$, hence by monotonicity of $\arccos(\cdot)$ on $[-1, 1]$,

$$d(m_y, m_j) = \arccos(m_y^\top m_j) \geq \arccos\bigl(\mu(\mathbf{M})\bigr). \tag{13}$$

Applying the triangle inequality to the triplet $(z, m_y, m_j)$ yields

$$d(z, m_j) \geq d(m_y, m_j) - d(z, m_y) \geq \arccos\bigl(\mu(\mathbf{M})\bigr) - d(z, m_y). \tag{14}$$

If $d(z, m_y) < \frac{1}{2} \arccos(\mu(\mathbf{M}))$, then the above inequality implies

$$d(z, m_j) > \frac{1}{2} \arccos\bigl(\mu(\mathbf{M})\bigr) > d(z, m_y), \tag{15}$$

and therefore $d(z, m_y) < d(z, m_j)$ holds for every $j \neq y$. This implies

$$m_y^\top z = \cos(d(z, m_y)) > \cos(d(z, m_j)) = m_j^\top z, \quad \forall j \neq y, \tag{16}$$

meaning that $m_y^\top z$ is the unique maximum among $\{m_k^\top z\}_{k=1}^K$ and thus $\hat{y}(z) = y$. Since the sufficient condition depends only on the global quantity $\mu(\mathbf{M})$, it holds uniformly over all classes $y$ and constitutes a worst-case angular margin guarantee.

The same guarantee can also be derived in a purely algebraic manner by decomposing $m_j$ and $z$ into components parallel and orthogonal to $m_y$ and applying Cauchy-Schwarz, avoiding the geodesic triangle inequality.

Moreover, the radius $\frac{1}{2}\arccos(\mu(\mathbf{M}))$ is worst-case tight w.r.t. $\mu(\mathbf{M})$: if a pair $(m_y, m_j)$ attains $m_y^\top m_j = \mu(\mathbf{M})$, then any point $z$ on the spherical bisector between them satisfy $\angle(z, m_y) = \angle(z, m_j) = \frac{1}{2}\arccos(\mu(\mathbf{M}))$. In this case, $z$ lies exactly on the decision boundary, implying that no uniform improvement is possible without additional structure assumptions beyond $\mu(\mathbf{M})$. $\square$

### A.2. Proof of Proposition 3.3

**Proposition 3.3 (Geometric Gradient Isolation).** *If the fixed output geometry satisfies inter-block orthogonality, i.e.,* $\mathbf{M}_{\mathrm{old}}^\top \mathbf{M}_{\mathrm{new}} = \mathbf{0}$*, then the gradient component driven by new tasks* $(\delta_{\mathrm{new}})$ *is automatically routed to the null space of the old task at the feature layer:*

$$\Pi_{\mathrm{old}}(\nabla_z \mathcal{L}) = \mathbf{M}_{\mathrm{old}}\delta_{\mathrm{old}}. \tag{17}$$

*Proof.* Let the unified anchor matrix be $\mathbf{M} = [\mathbf{M}_{\mathrm{old}}, \mathbf{M}_{\mathrm{new}}]$ and partition the logit-layer error as $\delta = [\delta_{\mathrm{old}}^\top, \delta_{\mathrm{new}}^\top]^\top$. By the chain rule for $\ell = \mathbf{M}^\top z$,

$$\nabla_z \mathcal{L} = \left(\frac{\partial \ell}{\partial z}\right)^\top \nabla_\ell \mathcal{L} = \mathbf{M}\delta = \mathbf{M}_{\mathrm{old}}\delta_{\mathrm{old}} + \mathbf{M}_{\mathrm{new}}\delta_{\mathrm{new}}. \tag{18}$$

Define the old anchor subspace $\mathcal{S}_{\mathrm{old}} \triangleq \mathrm{span}(\mathbf{M}_{\mathrm{old}})$, and let $\Pi_{\mathrm{old}}$ be the orthogonal projection onto $\mathcal{S}_{\mathrm{old}}$. We first show that $\mathrm{span}(\mathbf{M}_{\mathrm{new}}) \subseteq \mathcal{S}_{\mathrm{old}}^\perp$.

Take any vector $u \in \mathcal{S}_{\mathrm{old}}$. By definition of $\mathcal{S}_{\mathrm{old}}$, there exists $a$ such that $u = \mathbf{M}_{\mathrm{old}}a$. For any vector $b$, the inter-block orthogonality condition $\mathbf{M}_{\mathrm{old}}^\top \mathbf{M}_{\mathrm{new}} = \mathbf{0}$ yields

$$u^\top(\mathbf{M}_{\mathrm{new}}b) = a^\top \mathbf{M}_{\mathrm{old}}^\top \mathbf{M}_{\mathrm{new}}b = a^\top \mathbf{0}\, b = 0. \tag{19}$$

Hence every vector in $\mathrm{span}(\mathbf{M}_{\mathrm{new}})$ is orthogonal to every vector in $\mathcal{S}_{\mathrm{old}}$, i.e., $\mathrm{span}(\mathbf{M}_{\mathrm{new}}) \subseteq \mathcal{S}_{\mathrm{old}}^\perp$. In particular, $\mathbf{M}_{\mathrm{new}}\delta_{\mathrm{new}} \in \mathcal{S}_{\mathrm{old}}^\perp$, and thus

$$\Pi_{\mathrm{old}}(\mathbf{M}_{\mathrm{new}}\delta_{\mathrm{new}}) = 0. \tag{20}$$

Applying $\Pi_{\mathrm{old}}$ to (18) gives

$$\Pi_{\mathrm{old}}(\nabla_z \mathcal{L}) = \Pi_{\mathrm{old}}(\mathbf{M}_{\mathrm{old}}\delta_{\mathrm{old}}) + \underbrace{\Pi_{\mathrm{old}}(\mathbf{M}_{\mathrm{new}}\delta_{\mathrm{new}})}_{=0}. \tag{21}$$

The first term satisfies $\Pi_{\mathrm{old}}(\mathbf{M}_{\mathrm{old}}\delta_{\mathrm{old}}) = \mathbf{M}_{\mathrm{old}}\delta_{\mathrm{old}}$ since it lies in $\mathcal{S}_{\mathrm{old}}$, and the second term vanishes by (20). Therefore, $\Pi_{\mathrm{old}}(\nabla_z \mathcal{L}) = \mathbf{M}_{\mathrm{old}}\delta_{\mathrm{old}}$, as claimed.

$\square$

**Remark (Approximate orthogonality implies bounded leakage).** The exact orthogonality condition $\mathbf{M}_{\mathrm{old}}^\top \mathbf{M}_{\mathrm{new}} = \mathbf{0}$ can be relaxed to an $\varepsilon$-near-orthogonality. Specifically, if we define the proximity $\varepsilon \triangleq \sup_{v \in \mathcal{S}_{\mathrm{new}}, \|v\|_2 = 1} \|\Pi_{\mathrm{old}}v\|_2$, then by definition, it holds for any $g_{\mathrm{new}} \in \mathcal{S}_{\mathrm{new}}$ that:

$$\|\Pi_{\mathrm{old}}g_{\mathrm{new}}\|_2 \le \varepsilon \|g_{\mathrm{new}}\|_2. \tag{22}$$

This implies that perfect orthogonality is not strictly required; achieving approximate orthogonality is sufficient to bound the gradient leakage $\mathrm{Leak} \le \varepsilon$.

### A.3. Proof of Layer-wise Gradient Isolation under NC

This section proves that the source-level (logits→features) orthogonal isolation established in Proposition 3.3 propagates to earlier backbone layers under a (NC-consistent) spectral alignment condition. Throughout, we focus on the new-induced gradient component, i.e., the part driven by $\delta_{\text{new}}$.

Let the backbone produce the pre-normalization last-layer feature (immediately preceding the classifier) $h_L(x) \in \mathbb{R}^d$ and the normalized feature

$$z(x) \triangleq \frac{h_L(x)}{\|h_L(x)\|_2} \in \mathbb{S}^{d-1}. \tag{23}$$

For an intermediate layer representation $h_t(x) \in \mathbb{R}^{d_t}$ ($t = 0, 1, \ldots, L$), define the Jacobian

$$J_t(x) \triangleq \frac{\partial z(x)}{\partial h_t(x)} \in \mathbb{R}^{d \times d_t}, \qquad G_t(x) \triangleq J_t(x) J_t(x)^\top \in \mathbb{R}^{d \times d}. \tag{24}$$

To formalize layer-wise isolation, we define the old-sensitive subspace in the layer-$t$ activation space as the pullback of $\mathcal{S}_{\text{old}}$ through $J_t$:

$$\mathcal{S}_{\text{old}}^{(t)}(x) \triangleq \text{range}\big(J_t(x)^\top \Pi_{\text{old}}\big) \subseteq \mathbb{R}^{d_t}, \qquad \Pi_{\text{old}}^{(t)}(x) \triangleq \text{orthogonal projector onto } \mathcal{S}_{\text{old}}^{(t)}(x). \tag{25}$$

Thus, $\|\Pi_{\text{old}}^{(t)}(x) \nabla_{h_t} \mathcal{L}_{\text{new}}\|_2$ measures the leakage of new-induced gradients into old-sensitive directions at depth $t$.

Existing studies on neural collapse report that, in the late training stage, the last-layer geometry becomes highly structured and aligned, and empirical results show that NC-like structure can emerge across depth (Rangamani et al., 2023; Zhou et al., 2025). Motivated by these observations, we adopt the following NC-consistent non-mixing condition on the pullback metric.

**Assumption A.1** (NC-consistent subspace non-mixing (pullback metric))**.** For each layer $t$ and samples $x$, we assume the cross-subspace mixing of the pullback metric is small:

$$\big\|\Pi_{\text{old}} G_t(x) \Pi_{\text{new}}\big\|_2 \le \varepsilon_t, \qquad \big\|\Pi_{\text{new}} G_t(x) \Pi_{\text{old}}\big\|_2 \le \varepsilon_t, \tag{26}$$

for some (layer-dependent) scalar $\varepsilon_t \ge 0$. In the ideal NC-aligned limit, $\varepsilon_t = 0$.

**Lemma A.2** (Projector onto a matrix range)**.** *For any matrix $B \in \mathbb{R}^{m \times r}$, the orthogonal projector onto $\text{range}(B)$ is given by $\Pi_{\text{range}(B)} = B(B^\top B)^\dagger B^\top$, where $(\cdot)^\dagger$ is the Moore-Penrose pseudoinverse.*

**Proposition A.3** (Layer-wise propagation of gradient isolation)**.** *Under the assumption of inter-block orthogonality ($\mathbf{M}_{\text{old}}^\top \mathbf{M}_{\text{new}} = 0$) and Assumption A.1, for any layer $t$ and sample $x$, the gradient leakage is bounded by:*

$$\big\|\Pi_{\text{old}}^{(t)}(x) \nabla_{h_t} \mathcal{L}_{\text{new}}\big\|_2 \le \kappa_t(x) \varepsilon_t \big\|g_z^{\text{new}}\big\|_2, \tag{27}$$

*where $g_z^{\text{new}} = \mathbf{M}_{\text{new}} \delta_{\text{new}}$ is the feature-level error, and the conditioning factor is*

$$\kappa_t(x) \triangleq \big\|J_t(x)^\top\big\|_2 \cdot \big\|\big(\Pi_{\text{old}} G_t(x) \Pi_{\text{old}}\big)^\dagger\big\|_2. \tag{28}$$

*In particular, if $\varepsilon_t = 0$, then $\Pi_{\text{old}}^{(t)}(x) \nabla_{h_t} \mathcal{L}_{\text{new}} = 0$ for all $t$, i.e., the new-induced gradients remain orthogonal to all old-sensitive directions throughout backpropagation.*

*Proof.* By Proposition 3.3, the inter-block orthogonality $\mathbf{M}_{\text{old}}^\top \mathbf{M}_{\text{new}} = \mathbf{0}$ implies $\mathcal{S}_{\text{new}} = \text{span}(\mathbf{M}_{\text{new}}) \subseteq \mathcal{S}_{\text{old}}^\perp$. Therefore, the new-induced feature gradient $g_z^{\text{new}} = \mathbf{M}_{\text{new}} \delta_{\text{new}}$ satisfies $g_z^{\text{new}} = \Pi_{\text{new}} g_z^{\text{new}}$ and $\Pi_{\text{old}} g_z^{\text{new}} = 0$ (source isolation at the logits→features interface). We next show that under Assumption A.1, this isolation propagates to layer $t$ up to the mixing level $\varepsilon_t$.

Consider a fixed layer $t$ and sample $x$. By back-propagation:

$$\nabla_{h_t} \mathcal{L}_{\text{new}} = J_t(x)^\top \nabla_z \mathcal{L}_{\text{new}} = J_t(x)^\top g_z^{\text{new}}. \tag{29}$$

By definition, $\mathcal{S}_{\mathrm{old}}^{(t)}(x) = \mathrm{range}(J_t(x)^\top \Pi_{\mathrm{old}})$, hence applying Lemma A.2 with $B = J_t^\top \Pi_{\mathrm{old}}$ yields:

$$\Pi_{\mathrm{old}}^{(t)}(x) = \left(J_t^\top \Pi_{\mathrm{old}}\right)\left(\left(\Pi_{\mathrm{old}} J_t J_t^\top \Pi_{\mathrm{old}}\right)^\dagger\right)\left(\Pi_{\mathrm{old}} J_t\right)$$

$$= J_t^\top \Pi_{\mathrm{old}}\left(\Pi_{\mathrm{old}} G_t \Pi_{\mathrm{old}}\right)^\dagger \Pi_{\mathrm{old}} J_t. \tag{30}$$

Substituting this into the leakage term:

$$\Pi_{\mathrm{old}}^{(t)} \nabla_{h_t}\mathcal{L}_{\mathrm{new}} = J_t^\top \Pi_{\mathrm{old}}\left(\Pi_{\mathrm{old}} G_t \Pi_{\mathrm{old}}\right)^\dagger \Pi_{\mathrm{old}} J_t J_t^\top g_z^{\mathrm{new}}$$

$$= J_t^\top \Pi_{\mathrm{old}}\left(\Pi_{\mathrm{old}} G_t \Pi_{\mathrm{old}}\right)^\dagger \Pi_{\mathrm{old}} G_t\, g_z^{\mathrm{new}}. \tag{31}$$

Since $g_z^{\mathrm{new}} \in \mathcal{S}_{\mathrm{new}}$, we have $g_z^{\mathrm{new}} = \Pi_{\mathrm{new}} g_z^{\mathrm{new}}$, hence

$$\Pi_{\mathrm{old}} G_t\, g_z^{\mathrm{new}} = \Pi_{\mathrm{old}} G_t \Pi_{\mathrm{new}}\, g_z^{\mathrm{new}}. \tag{32}$$

Taking operator norms and using submultiplicativity yields

$$\left\|\Pi_{\mathrm{old}}^{(t)} \nabla_{h_t}\mathcal{L}_{\mathrm{new}}\right\|_2 \le \|J_t^\top\|_2 \cdot \left\|\left(\Pi_{\mathrm{old}} G_t \Pi_{\mathrm{old}}\right)^\dagger\right\|_2 \cdot \left\|\Pi_{\mathrm{old}} G_t \Pi_{\mathrm{new}}\right\|_2 \cdot \|g_z^{\mathrm{new}}\|_2$$

$$\le \kappa_t(x)\,\varepsilon_t\,\|g_z^{\mathrm{new}}\|_2, \tag{33}$$

which proves (27). If $\varepsilon_t = 0$, then $\Pi_{\mathrm{old}} G_t \Pi_{\mathrm{new}} = 0$ and the right-hand side is zero, implying $\Pi_{\mathrm{old}}^{(t)} \nabla_{h_t}\mathcal{L}_{\mathrm{new}} = 0$ for all $t$.

$\square$

**Remark 1 (Interpretation and conditioning).** The bound in Eq. (27) separates the leakage into two factors: the alignment error $\varepsilon_t$ and the geometric conditioning $\kappa_t(x)$. Mathematically, the term $\|(\Pi_{\mathrm{old}} G_t \Pi_{\mathrm{old}})^\dagger\|_2$ equals $1/\lambda_{\mathrm{min}}^+$, i.e., the inverse of the smallest non-zero eigenvalue of the old-task pullback metric. Thus, a large $\kappa_t(x)$ (or small $\lambda_{\mathrm{min}}^+$) indicates that the old-task pullback geometry becomes nearly degenerate, in which case the bound can be conservative. This insight motivates our Level-3 alignment (Subspace-Geometric Retention): by explicitly anchoring high-reliability old-class samples via Eq. (10), we actively prevent such geometric degeneration, thereby improving the conditioning of the bound in practice.

**Remark 2 (SF-OW setting).** We present this analysis primarily as a motivation to impose structural constraints from the output interface. It should be noted that isolation at the logits-to-feature interface purely from the fixed head geometry in Proposition 3.3 is strict, while the layer-wise propagation result in Proposition A.3 is conditional: it holds when the pullback metric remains approximately block-diagonal with respect to the anchor-induced old/new decomposition (Assumption A.1).

In the SF-OW setting, while unsupervised adaptation may theoretically risk violating this condition, our framework mitigates this via (i) a fixed classifier head which anchors the discriminative subspace, and (ii) explicit retention terms (Level-3 in TLGA) that stabilize old-class directions. Empirically, we report a low cross-task interference metric (Leak) in Table 2, which is consistent with limited cross-task leakage during adaptation.

## B. Experimental Setup

### B.1. Datasets and processing

We evaluate TLGA on three public RFFI benchmarks covering different wireless communication protocols and acquisition setups, each reflecting a different challenge in open-world RFFI: a widely adopted standard benchmark (WiSig), fine-grained hardware distinction (Oracle), and large-scale category scalability (LoRa). Detailed statistics and the exact preprocessing configurations used in this paper are summarized in Table 3.

*Table 3.* **Dataset statistics and preprocessing configurations.** *Setting* denotes the specific experimental configuration used. $L_{IQ}$ represents the length of complex IQ segments input to the preprocessing pipeline. The Split column indicates the ratio of old ($K_{old}$) to new ($K_{new}$) classes. All network inputs are normalized and stacked into 3 channels.

| Dataset | Setting | Protocol | Segment | $K$ | Split ($K_{old} : K_{new}$) | $N_{total}$ | $L_{IQ}$ | Net Input ($H \times W \times C$) |
|---------|---------|----------|---------|-----|------------------------------|-------------|----------|-----------------------------------|
| WiSig | ManySig | WiFi 802.11a/g | Preamble | 6 | 4 : 2 | 24,000 | 256 | $32 \times 32 \times 3$ |
| Oracle | 62ft | WiFi 802.11a | Raw IQ | 15 | 10 : 5 | 15,000 | 6000 | $32 \times 32 \times 3$ |
| LoRa | - | LoRa | Preamble | 25 | 20 : 5 | 25,000 | 8192 | $102 \times 62 \times 3$ |

- **WiSig (Hanna et al., 2022).** WiSig is collected on the ORBIT testbed and is widely recognized as the foundation benchmark for RFFI due to its high signal quality, large scale, and comprehensive coverage across multiple days and receivers. It is commonly used as a standard to validate general model performance. We select the *ManySig* subset, which contains sufficient per-device samples collected from 6 identical Atheros 5212 chipsets as transmitters recorded by 12 USRP B210 and N210 receivers over 4 days. Following common practice in RFF studies, we extract a fixed preamble segment for fingerprinting.

- **Oracle (Sankhe et al., 2019).** Oracle targets the challenging bit-similar device identification task, where transmitters share identical hardware configurations and transmit IEEE 802.11a frames. The dataset contains recordings under multiple Tx-Rx distance settings. We use the 62ft setting, where signals from 15 USRP X310 transmitters are received by a single USRP B210. Given the proven efficacy of raw IQ features for this dataset, we directly partition the provided raw signal streams into fixed-length frames without additional preamble extraction.

- **LoRa (Shen et al., 2022).** LoRa represents large-scale IoT device identification with multi-model, off-the-shelf commercial LoRa devices captured by a USRP N210. To the best of our knowledge, it features the largest number of distinct transmitter classes among public RFF datasets. We use a 25-device set and extract the payload-independent preamble segments for classification.

For each dataset, we apply a class-stratified random split (50%/50%) into labeled source $\mathcal{D}_s$ (used for Stage 1 pre-training) and unlabeled mixed old/new target $\mathcal{D}_t$ (used for Stage 2 adaptation). This follows the transductive standard protocol commonly adopted in prior RFFI and OW evaluation settings. On high-SNR benchmarks, the induced $\mathcal{D}_s \to \mathcal{D}_t$ shift can be relatively mild, so some metrics may approach a ceiling.

To mitigate channel effects, raw IQ sequences are converted to channel-invariant log-power spectrograms. Specifically, we apply Short-Time Fourier Transform (STFT) using a Hamming window of size $N_{FFT} = 128$, with an overlap of 50%. We compute the squared magnitude to obtain the power spectrogram $S(f, t)$ and apply adjacent-frame ratioing $S(:, t)/S(:, t-1)$ to cancel out slow-varying channel components, followed by logarithmic scaling and normalization.

Figure 7 illustrates the preprocessed channel-invariant spectrograms from four distinct transmitters of the same model from the LoRa dataset. Although these devices follow the same protocol and exhibit nearly identical macro-level patterns, they possess minute, hardware-intrinsic differences. The subtlety of these variations highlights the inherent challenge of fine-grained RFFI in SF-OW setting.

### B.2. Evaluation Metrics

Following established protocols (Cao et al., 2021; Xiao et al., 2024), we evaluate the model performance in a transductive setting on the unlabeled target stream. We employ three accuracy-based metrics, one trade-off metric, and one separability metric:

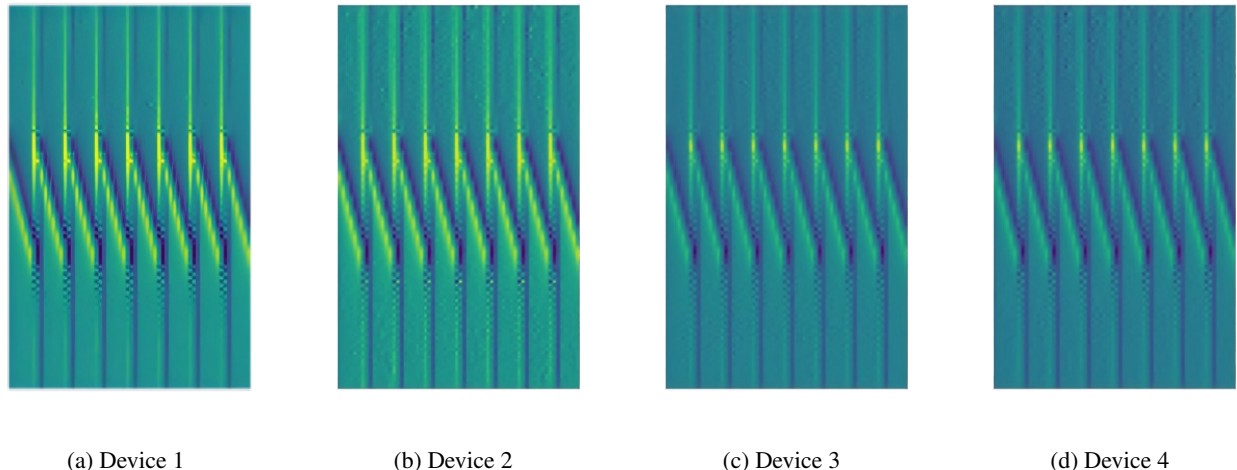

| (a) Device 1 | (b) Device 2 | (c) Device 3 | (d) Device 4 |

*Figure 7.* Visualization of channel-invariant spectrograms from four transmitters (LoRa).

- **Accuracies.** Old Accuracy assesses stability by measuring the retention of source-trained knowledge on old devices. New Accuracy and All Accuracy evaluate plasticity and global performance. Since new classes lack prior labels, we solve the linear assignment problem via the Hungarian algorithm to establish an optimal permutation $\mathrm{perm}(\cdot)$ between predicted clusters and ground truth for $\mathcal{Y}_{\mathrm{new}}$ and $\mathcal{Y}_{\mathrm{total}}$, computing accuracy $\frac{1}{N}\sum_i \mathbb{1}[y_i = \mathrm{perm}(\hat{y}_i)]$.

- **H-score.** The H-score is the harmonic mean of Old Accuracy ($\mathcal{A}_{\mathrm{old}}$) and New Accuracy ($\mathcal{A}_{\mathrm{new}}$), calculated as $H = \frac{2 \cdot \mathcal{A}_{\mathrm{old}} \cdot \mathcal{A}_{\mathrm{new}}}{\mathcal{A}_{\mathrm{old}} + \mathcal{A}_{\mathrm{new}}}$. This metric is critical for SF-OW RFFI task as it penalizes models that favor one task while neglecting the other, serving as a robust measure of the trade-off between stability and plasticity (Saito & Saenko, 2021).

- **AUROC.** To quantify the model's capability in open-set recognition (i.e., distinguishing old from new transmitters), we report the AUROC (Fu et al., 2020), utilizing the maximum old-class probability $s(x) = \max_{c \in \mathcal{Y}_{\mathrm{old}}} p_\theta(c \mid x)$ as the decision statistic. A high AUROC reveals that the feature manifolds of old and new devices are well-separated.

## B.3. Baseline Configurations

To systematically evaluate the effectiveness of our SF-OW RFFI framework, we benchmark against a comprehensive set of representative state-of-the-art baselines that are widely used in recent related literature (Xiao et al., 2024; Han et al., 2025). Our selection is organized around two dominant paradigms for addressing the stability-plasticity trade-off:

- **Continual Learning (CL)**: methods that primarily target catastrophic forgetting (covering distillation, parameter-importance regularization, and gradient/subspace projection).

- **Open-World (OW)**: recognition methods that emphasize joint old-class recognition and multi-new separation, typically employing advanced clustering and bias-correction mechanisms to isolate new distributions.

Below is a brief introduction to each baseline:

**CL baselines.** We benchmark against representative CL methods that cover the primary paradigms for mitigating catastrophic forgetting:

- **LwF** (Li & Hoiem, 2017) (*distillation-based CL*). A seminal CL baseline that preserves old knowledge via functional regularization: it distills the previous model's outputs on old classes while learning on new data. By incorporating a distillation loss term computed on the new samples, it discourages the model from drifting away from the representations learned in previous stages.

- **EWC** (Kirkpatrick et al., 2017) (*Fisher-based regularization*). A weight-importance regularizer that models forgetting as harmful drift along parameters critical to past tasks, quantified by a diagonal Fisher approximation. EWC remains a standard representative of importance-weighted stability constraints and is frequently used as a strong CL reference across domains.

- **MAS** (Aljundi et al., 2018) (*sensitivity-based regularization*). MAS estimates parameter importance by measuring the sensitivity of the network outputs to parameter perturbations, yielding a source-agnostic alternative to Fisher-based scores. Its key highlight is that it regularizes the model through function sensitivity rather than likelihood curvature, often providing more robust importance estimation under distribution shifts.

- **GPM** (Saha et al., 2021) (*gradient projection*). A representative gradient-control method that identifies the core subspace of past tasks via SVD on feature representations. By projecting new gradient updates onto the null space of this subspace, it ensures that learning new tasks does not interfere with the feature directions critical for previous knowledge.

**OW baselines.** We include representative OW/category-discovery baselines that jointly address old-class recognition and multi-new separation, often through discovery objectives and debiasing/refinement mechanisms:

- **SHOT** (Liang et al., 2020) (*canonical source-free adaptation*). A widely adopted SFDA baseline that freezes the source classifier and adapts only the feature extractor on target data, typically combining information maximization with pseudo-label self-training to refine representations under strict source-free constraints.

- **ORCA** (Cao et al., 2021) (*uncertainty-aware OW recognition*). An influential OW framework that tackles old-class bias by using uncertainty-based gating to down-weight ambiguous samples and then applying iterative refinement (e.g., consistency/contrastive-style signals) to jointly maintain old recognition while progressively structuring unknowns.

- **OpenLDN** (Rizve et al., 2022) (*discovery with refinement*). A neighborhood-driven framework that strengthens novel discovery in mixed data through a refinement loop, repeatedly cleaning pseudo labels to reduce assignment collapse and prevent novel clusters from being swallowed by dominant known categories.

- **GCD** (Vaze et al., 2022) (*generalized category discovery*). A representative mixed-label formulation that couples supervised known-class classification with unsupervised novel discovery in a unified space, typically using consistency-style objectives so that novel categories can emerge without being biased toward the labeled known set.

- **OpenNCD** (Liu et al., 2023a) (*new-class discovery*). A representative NCD baseline that discovers multiple unseen classes from unlabeled data via prototype-driven learning, where prototypes and their assignments are progressively refined to stabilize training against noisy pseudo labels and improve multi-novel separation.

**Unified SF-OW protocol.** Since most CL and OW methods are not natively designed for the Source-Free Open-World setting (no source data + unsupervised multi-new discovery). To ensure fairness and reproducibility, we adapt all baselines under the same constraints: (i) no source samples are stored or replayed; (ii) all methods start from the same Stage 1 initialization; and (iii) all methods are evaluated on the same target stream with $K_{\text{new}}$ assumed known:

- **Teacher (Reference).** Teacher denotes the frozen Stage 1 model trained on source classes only, without any target adaptation. Since it has no new-class decision slots, we report its New performance by k-means clustering ($K_{\text{new}}$ clusters) in the teacher feature space and aligning cluster IDs to ground-truth new labels via Hungarian matching; Old accuracy is obtained from the original old-class head.

- **CL-based adaptation.** Standard CL baselines typically assume supervised labels for new tasks/classes and do not directly provide an open-world discovery mechanism. To make them applicable in SF-OW, we attach all CL baselines to a common self-training (ST) pipeline that provides pseudo supervision on the unlabeled target stream, yielding **ST Only** and **ST + CL** variants. The corresponding CL regularizers (e.g., distillation, parameter importance, or gradient projection) are then applied on top of ST to mitigate forgetting of old classes.

- **OW-based adaptation.** Most OW or category-discovery methods are semi-supervised and rely on labeled source data to anchor old classes. Under strict source-free constraints, we replace source supervision with a teacher-guided

pseudo-source constructed from high-confidence target samples predicted as known by the frozen Stage 1 teacher, while the remaining target samples are treated as unlabeled for new discovery. All OW baselines are then run following their original objectives under this pseudo-source setting, and New/All metrics are computed with Hungarian matching for comparability.

### B.4. Implementation Details

**Hardware and software.**   All experiments and evaluations are implemented on a server equipped with a 13th Gen Intel(R) Core(TM) i9-13900K CPU, NVIDIA GeForce RTX 4090 GPU, and 128GB of RAM. Our proposed SF-OW RFFI method and baselines are implemented in Python 3.8.20, with models built and trained using PyTorch 2.4.1.

**Training schedule.**   For fair comparison, we employ a uniform ResNet-18 backbone with a feature dimension of $d = 512$ for all experiments. The model is trained for 100 epochs with a 15-epoch warm-up, employing the Adam optimizer with a batch size of 200 and an initial learning rate of $5 \times 10^{-4}$ which decays via a cosine scheduler. To stabilize pseudo-label generation, we maintain an EMA teacher with a momentum of 0.95. For the geometric ablation, we constrain the feature dimension to $d = 32$ to accentuate the structural effects within a tighter feature space, as the intrinsic quasi-orthogonality of high-dimensional spaces naturally trivializes separation and masks the impact of explicit geometric designs.

**Hyperparameter settings.**   For IO-ETF optimization in Eq. (4), we initialize $\mathbf{M}_{\text{new}}$ randomly and optimize it prior to adaptation using Riemannian Adam on the unit hypersphere to enforce the norm constraint. We set the smoothness parameter $\alpha = 20$ and the balancing weight $\lambda = 1.0$. In BEOT, the Sinkhorn-Knopp algorithm is configured with $\epsilon = 0.05$ with 3 iterations. For PMA, the feature bank stores momentum features for the full dataset (used as the denominator in Eq. (6)), and the evolutionary weight $w(t)$ follows a linear ramp-up across training epochs, and the temperature $\tau$ is set to 0.5. For RAPR, we employ a confidence threshold $\gamma = 0.6$ and a retention ratio $\kappa = 0.5$ (top-50%) for filtering, with the retention loss weighted by $\lambda_{\text{old}} = 20.0$.

**Remark.**   Note that the adaptation stage training overhead is operationally negligible compared to the long intervals of new device arrivals in typical wireless network scenarios. Furthermore, the inference process remains identical to standard baselines under the same ResNet classifier, incurring zero additional latency and preserving the rapid response capabilities required for real-time RFFI deployments.

# C. Detailed Experimental Results

## C.1. Failure Case Analysis

Figure 8 shows the confusion matrix of our method on LoRA dataset. Overall, our method exhibits no global catastrophic forgetting or classification collapse. The near-perfect diagonal among old classes confirms historical knowledge is well preserved. Two specific failure modes emerge:

- Asymmetric old-new confusion: The clearest case is N23→O6 (20%), while the reverse drift is significantly weaker (9% O6→N23). This asymmetry indicates RAPR successfully shields old-class identities against new stream interference.

- Hard new-new confusion: We also observe a specific confusing new pair, N21↔N22 ( 10%), while almost none exists among remaining pairs. This isolated error indicates RFFs of these two hard new classes remain too similar to be cleanly separated, especially for unlabeled mixed streams.

Therefore, the dominant remaining failures are not global, but localized ambiguities for a few hard new classes and borderline old classes. These failures are mainly attributed to the inherent physical hardware similarity, rather than algorithmic instability.

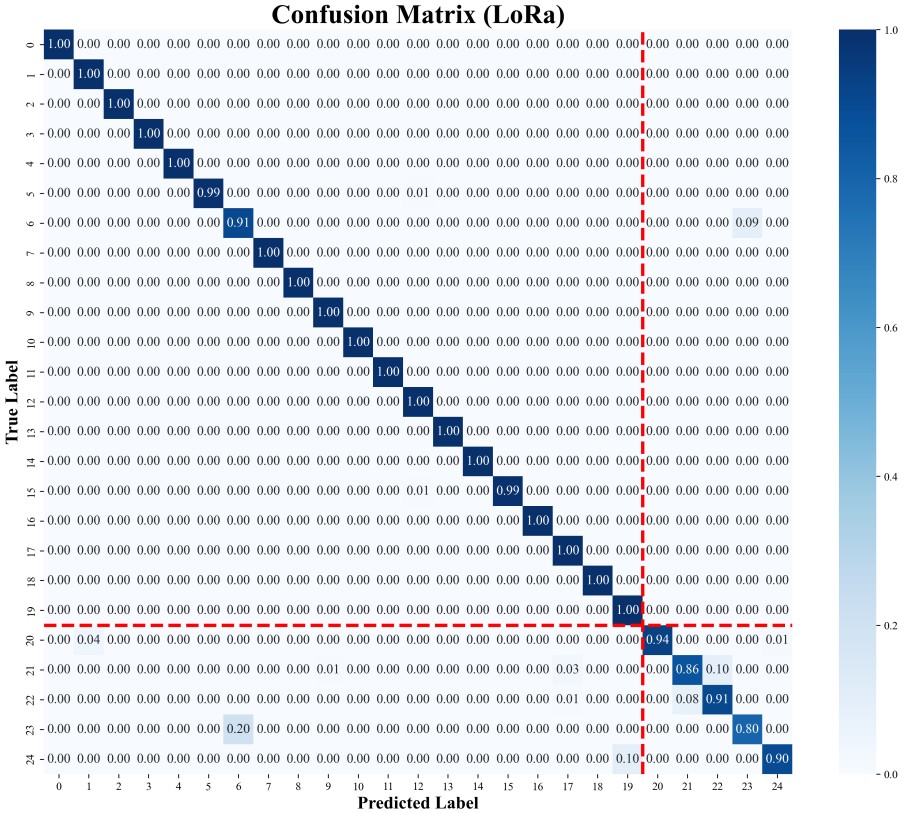

*Figure 8.* Confusion matrix of our method.

## C.2. Ablation on TLGA

**Figure 9 details the ablation impact on Old/New/All accuracies and AUROC across three datasets.** We analyze the specific role of each TLGA component as follows.

**w/o BEOT.** Removing semantic registration BEOT causes the dominant failure on new discovery. On Oracle, New Acc drops sharply from 99.65% to 40.41% (All Acc 99.88% → 66.67%), confirming that the capacity-constrained global assignment is essential to prevent pseudo-label collapse and to activate dormant new slots.

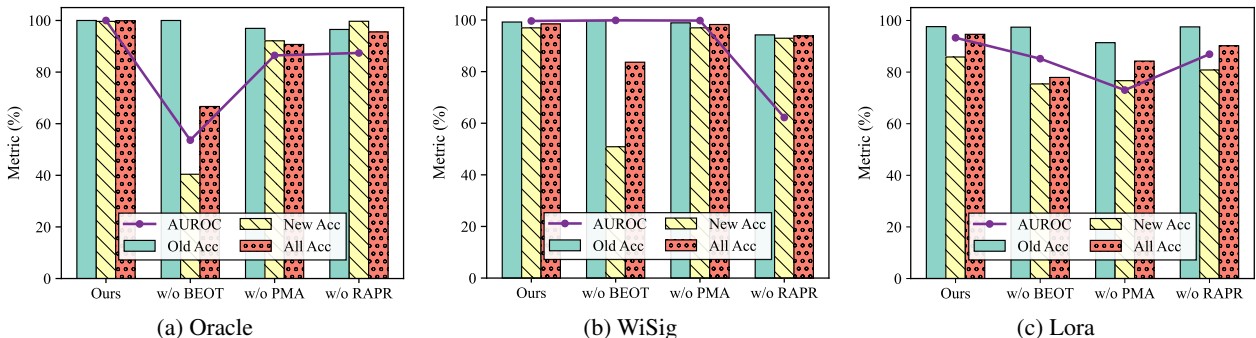

*Figure 9.* Ablation on three-level geometric alignment.

**w/o PMA.** Figure 9 also shows that the absence of PMA consistently degrades performance across all metrics, with the impact being most pronounced on the challenging LoRa dataset, where all performance metrics drop by approximately 15%. This sharp decline suggests that forcibly aligning loose manifolds to rigid anchors without soft-to-hard topological guidance tears local neighborhood structures, causing simultaneous regression in both stability and plasticity.

**w/o RAPR.** Its removal triggers a catastrophic drop in open-set separability. The AUROC score on Oracle degrades by around 40%, from near 100% to approximately 60%. This indicates that RAPR effectively pins the old-class subspace; without it, known features drift towards the new subspace, blurring the decision boundary between old and new distributions, severely compromising the model's ability to detect new transmitters.

These results validate that the three levels of TLGA are mutually reinforcing necessities, each addressing a distinct alignment failure mode. BEOT prevents semantic assignment collapse, PMA averts manifold topological tearing, and RAPR suppresses subspace boundary drift. **The absence of any level breaks this protective hierarchy, confirming that a holistic semantic-manifold-subspace alignment is prerequisite for robust SF-OW adaptation.**

### C.3. Scalability

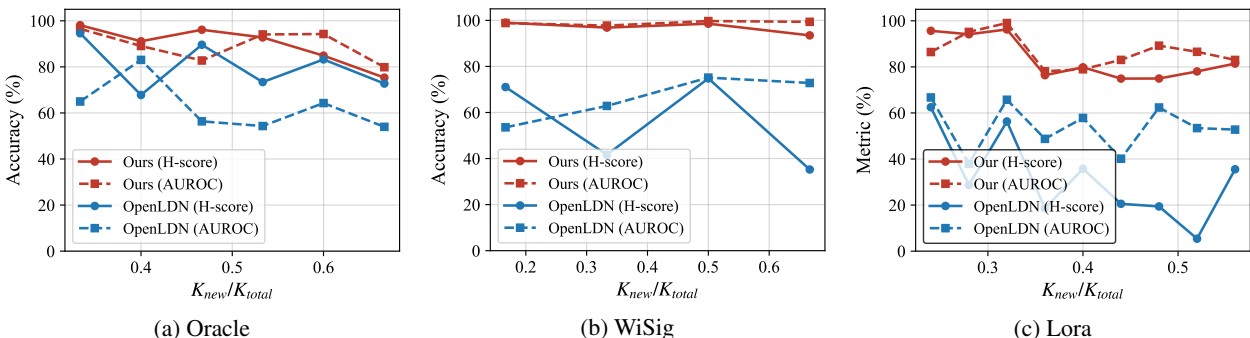

*Figure 10.* Model scalability to varying new class ratios.

**Figure 10 provides a stress test of model scalability by sweeping the new-class ratio ($K_{\text{new}}/K_{\text{total}}$) from mild to openness across three benchmarks.** We compare the H-score (stability-plasticity balance) and AUROC (open-set separability) against the strongest baseline.

As the ratio of new classes increases, the task complexity grows exponentially due to the expanded search space for unsupervised clustering. Baseline methods exhibit severe degradation. For instance, on LoRa, as shown in Figure 10 (c), SF-OpenLDN's H-score collapses to near zero under high openness. This failure stems from the instability of self-supervised clustering when the majority of data belongs to unknown categories. **In contrast, our method maintains robust resilience.** Even in the most challenging regime (e.g., LoRa with $> 50\%$ novel classes), we sustain an H-score and AUROC exceeding 80%.

**The superior scalability is inherent to the IO-ETF framework.** Unlike baselines that must dynamically discover cluster centers in a noisy high-dimensional space, our approach effectively reduces the problem to filling pre-fixed, maximally separated geometric slots. This slot-filling paradigm is inherently more robust to the challenges of dimensionality and class imbalance than pure discovery-based methods, ensuring stable performance across varying new-class loads.

