# OpenReview forum: "Source-Free Open-World RF Fingerprint Identification"
_ICML.cc/2026/Conference — ICML 2026 regular_

### Official Review · Reviewer_e42A · 2026-02-27

**Soundness:** 3
**Presentation:** 3
**Significance:** 3
**Originality:** 3
**Overall Recommendation:** 4
**Confidence:** 4

**Summary:**

This paper tackles the novel and challenging problem of Source-Free Open-World Radio Frequency Fingerprint Identification (SF-OW RFFI), where a model pre-trained on known devices must adapt to an unlabeled stream containing both known and unknown devices, without access to the original source data. The authors propose a two-stage solution: (1) Incremental Orthogonal Equiangular Tight Frame (IO-ETF) to construct geometrically optimal classifier heads that maximize intra-task separability and inter-task orthogonality; and (2) Triple-Level Geometric Alignment (TLGA) to align unlabeled data to this fixed geometry via balanced assignment, progressive anchoring, and reliability-aware pseudo-replay. Experiments on three RFF benchmarks demonstrate strong performance in old-class retention and new-class discovery.

**Compliance With Llm Reviewing Policy:**

Affirmed.

**Key Questions For Authors:**

Please refer to the comments.

**Limitations:**

yes

**Strengths And Weaknesses:**

Strengths

1. Problem Novelty and Relevance: The SF-OW setting is highly relevant for real-world deployment, where privacy and storage constraints prevent source data access, and new devices appear dynamically.
2. Strong Theoretical Foundation: The paper provides a rigorous motivation based on neural collapse and gradient isolation, showing how fixed ETF geometry can simultaneously enhance separability and prevent forgetting.
3. Well-Designed Framework: The IO-ETF construction and TLGA are modular, well-motivated, and technically sound. The three levels of alignment (global, local, and subspace) address different aspects of the stability-plasticity trade-off.
4. Comprehensive Evaluation: Experiments cover three diverse RFF datasets。

Weaknesses

1. Complexity and Overhead: The framework introduces multiple components with several hyperparameters. While ablation shows each contributes, the overall complexity may hinder adoption in resource-constrained or real-time RFFI systems.
2. Assumption of Known New Class Count: The approach assumes the number of new classes is known. Although sensitivity analysis is provided, practical scenarios often lack this knowledge, and the method's performance degrades noticeably with underestimation.
3. Limited Analysis of Failure Cases: The paper reports strong average performance but does not deeply analyze when and why the method fails—e.g., which new classes are confused, or which old classes are forgotten. A confusion matrix or per-class analysis would strengthen the evaluation.

---

> ### Author Rebuttal · Authors · 2026-03-31
>
> Dear Reviewer e42A,
>
> We thank the reviewer for the thorough review and valuable insights. We provide responses to your concerns below.
>
> >**W1. Complexity and Deployment Considerations**
>
> To clarify the complexity, we explicitly decouple the adaptation (training) and deployment (inference) phases.
>
> 1. Efficient deployment: In real-world RFFI systems, real-time and resource constraints mainly arise during deployment phase (i.e., how efficiently the deployed model can identify active devices), rather than occasional backend adaptation.
> Our final inference path remains a standard forward pass as the baselines. Profiling shows a lightweight model with 13.02M parameters (~49.7 MB) and 0.56 GFLOPs per sample. The inference latency is 0.039 ms, yielding a >25,000 samples/s throughput. This indicates that our framework is well suited to real-time and resource-constrained RFFI deployment.
>
> 2. Manageable adaptation: We quantify adaptation complexity under a unified protocol by measuring one-time initialization, per-epoch runtime, and peak GPU memory.
>
> |Method|ST Only|ST+EWC|SF-GCD|SF-OpenLDN|**Ours**|
> |-|-|-|-|-|-|
> |Init. Time(s)|4.80|6.12|3.96|3.20|4.26|
> |Time/Ep.(s)|1.05|1.10|1.73|1.74|2.12|
> |GPU Memory(MB)|2930|3019|3828|3835|2712|
> |H-score(%)|90.13|90.22|89.33|88.46|97.05|
>
> Results show that our method achieves a markedly better stability-plasticity trade-off (>7% H-score gain) at a manageable cost. TLGA mainly uses batch-level matrix operations, while fixing the classifier anchors avoids extra gradient/optimizer states for the classifier head, which keeps memory low. Moreover, IO-ETF adds only an easy one-time setup, and per-epoch runtime increases by only ~1s over the simplest baseline.
>
> Deployment Considerations: **Overall, our framework yields a solid performance gain with manageable training cost and can be efficiently deployed in practical RFFI systems.** For inference, our lightweight model supports strict real-time recognition. Meanwhile, adaptation does not constitute a bottleneck in practice. As the emergence of new devices is typically discrete and sporadic, the computational overhead for adaptation is amortized over time. And it usually runs on resource-abundant access points or edge servers, while constrained end-devices are only responsible for signal transmission.
>
> >**W2. Known $K_{new}$ Assumption**
>
> While assuming a known $K_{new}$ is a common OW/GCD simplification to isolate variables and focus on core representation learning, we fully agree that practical deployments may lack this prior.
>
> As you insightfully observed, our geometric framework exhibits an asymmetric tolerance to $K_{new}$ errors, making it less sensitive to over-estimation (App. C.4). Practically, lightweight clustering estimators (e.g., DPMM and Silhouette-selected K-means) can be employed beforehand to infer an initial class count.
>
> |Dataset(K)|Estimator|$\hat{K}$|Acc(%)|AUROC(%)|
> |-|-|-|-|-|
> |WiSig(6)|DPMM|6|99.96|99.73|
> ||K-means|6|99.96|99.73|
> |Oracle(15)|DPMM|19|90.37|94.59|
> ||K-means|14|86.52|94.49|
> |LoRa(25)|DPMM|23|91.71|91.97|
> ||K-means|25|94.64|93.38|
>
> Results show that coarse estimates are already sufficient to maintain competitive recognition performance. Further leveraging the asymmetry, we can strategically reserve an upper margin, allowing the skeleton to pre-allocate redundant anchors for potential new classes, ultimately keeping stable overall performance.
>
> >**W3. In-Depth Analysis of Failure Cases**
>
> To perform a fine-grained failure-case analysis, we plotted the confusion matrix (https://anonymous.4open.science/r/cm_RFFI/RFFI_cm.png) and conducted a per-class diagnosis.
>
> |ID|Acc|Top Error|
> |-|-|-|
> |O6|0.91|N23(0.09)|
> |N21|0.86|N22(0.10)|
> |N22|0.91|N21(0.08)|
> |N23|0.80|O6(0.20)|
>
> Overall, our method exhibits no global catastrophic forgetting or classification collapse. The near-perfect diagonal among old classes confirms historical knowledge is excellently preserved. Two specific failure modes emerge:
>
> 1. Asymmetric old-new confusion: The clearest case is N23→O6 (20%), while the reverse drift is significantly weaker (9% O6→N23). This asymmetry proves RAPR successfully shields old-class identities against new stream interference.
>
> 2. Hard new-new confusion: We also observe a specific confusing new pair, N21↔N22 (~10%), while almost none exists among remaining pairs. This isolated error indicates RFFs of these two hard new classes remain too similar to be cleanly separated, especially for unlabeled mixed streams.
>
> Therefore, the dominant remaining failures are not global, but localized ambiguities for a few hard new classes and borderline old classes. These failures are fundamentally bounded by the inherent physical hardware similarity, rather than algorithmic instability.
>
> ***
> Thanks so much for your support and insightful suggestions for improving our work. We will make sure to add more discussion regarding these points in the revision, and try our best to address any remaining concerns you might have.

---

> > ### Author Rebuttal · Reviewer_e42A · 2026-04-04
> >
> > Thank you for the rebuttal.

---

> > > ### Author Response · Authors · 2026-04-06
> > >
> > > Dear Reviewer e42A,
> > >
> > > Thank you so much for taking the time to review our response and for officially confirming that our rebuttal has fully resolved your concerns!
> > >
> > > We deeply appreciate your highly constructive and professional suggestions throughout this process, which have helped us improve our work. Please do not hesitate to let us know if any further information is needed.
> > >
> > > Thank you again for your valuable time and your strong support! We sincerely hope our efforts could earn your further recognition.

---

### Official Review · Reviewer_B73a · 2026-03-12

**Soundness:** 3
**Presentation:** 2
**Significance:** 3
**Originality:** 3
**Overall Recommendation:** 4
**Confidence:** 1

**Summary:**

This paper studies source-free open-world radio-frequency fingerprint identification (RFFI), a task in which a model trained on a set of known transmitter devices must recognize signals from known devices while also detecting previously unseen ones, without access to source data during adaptation. The authors propose a framework that combines prototype learning, clustering, and unknown-class detection mechanisms to address open-world conditions in RF environments. The method is evaluated on benchmark datasets for RF fingerprinting and is compared with existing approaches.

**Compliance With Llm Reviewing Policy:**

Affirmed.

**Final Justification:**

The paper addresses an interesting and relevant problem and presents a method with strong empirical results.

My main concern was the clarity and accessibility of the overall pipeline. The authors’ rebuttal and follow-up response significantly improve this aspect by providing a clearer and more structured explanation of the two-stage design and the interaction between components.

Given my limited domain expertise, I rely primarily on the empirical evaluation and high-level assessment, which suggest that the approach is technically sound and effective. Overall, my concerns have been largely addressed, and I maintain my positive evaluation.

**Key Questions For Authors:**

1. Could the authors provide more high-level intuition for the overall pipeline? In particular, it would be helpful to understand how the different modules (e.g., prototype learning, clustering, and unknown-class detection) interact and which component is most critical for performance.
2. The paper studies a source-free open-world setting. What assumptions are made about the distribution shift between training and deployment environments? How sensitive is the method to violations of these assumptions?
3. How does the proposed approach compare conceptually to existing open-set or open-world recognition methods outside the RF domain? A clearer positioning relative to broader literature would help readers understand the novelty of the contribution.
4. The experiments demonstrate improvements on specific datasets, but it is unclear how well the method generalizes to other RF conditions, hardware variations, or noise levels. Could the authors comment on the robustness of the approach to different deployment scenarios?
5. Could the authors clarify the computational complexity and practical deployment requirements of the method, particularly in real-time or resource-constrained environments?

**Limitations:**

No. The paper focuses on a specific RF fingerprinting scenario with particular datasets and experimental conditions. It remains unclear how well the approach generalizes to other RF environments, hardware conditions, or signal types. A discussion of practical limitations and deployment considerations would strengthen the paper.

**Strengths And Weaknesses:**

This work falls outside my main area of expertise, and therefore my evaluation is necessarily at a relatively high level rather than an in-depth technical assessment. From a high-level perspective, the paper addresses a relevant problem in RF signal processing and machine learning, namely identification under open-world conditions without source data. The proposed approach appears to combine several components designed to improve robustness to unseen devices, and the experimental section suggests that the method performs competitively on the evaluated datasets.

However, the paper is difficult to follow for readers outside this specific subfield. The presentation relies heavily on domain-specific terminology from RF signal processing, and several methodological components are introduced with limited intuition. In particular, the motivation behind some design choices and algorithmic steps is not always clear, which makes it challenging to understand how the different parts of the framework interact. Additional high-level explanations and clearer descriptions of the pipeline would improve accessibility.

Furthermore, some definitions and notations are introduced without sufficient context, and the relationship between the proposed method and prior work could be clarified more explicitly. While the experimental results appear extensive, it is difficult to assess the technical novelty and correctness of the approach without deeper familiarity with the domain. In particular, I did not carefully review the technical proofs and therefore cannot assess their correctness.

---

> ### Author Rebuttal · Authors · 2026-03-31
>
> Dear Reviewer B73a,
>
> Thank you for your time and expertise in reviewing our submission. Please find our responses below.
> >**Q1. High-Level Intuition**
>
> At a high level, our SF-OW framework operates on a "**pre-defined anchor, stable alignment**" strategy in feature space.
>
> IO-ETF (prototype construction) determines *where class centers should be placed*. It constructs an optimal geometry for SF-OW classifier-head anchors, enforcing maximal new-class separation and explicit old-new isolation.
>
> TLGA (geometry-guided clustering) dictates *how features align* towards these anchors stably with three modules working sequentially across semantic, manifold, and subspace levels:
> * BEOT determines the target anchor for each sample (*where to go*);
> * PMA drives features toward anchors while preserving local topology (*how to go*);
> * RAPR locks reliable old-class cores, preventing subspace drift during alignment (*what to guard*).
>
> Unknown detection is not a standalone module, but emerges naturally from matching target features to IO-ETF anchors.
>
> Remark: IO-ETF is necessary as it provides the structural prior preventing blind self-clustering for the whole framework. Within TLGA, criticality splits by objective (Figs.5&9): BEOT is vital for new-class discovery, while RAPR directly dictates old-class retention.
> >**Q2. Assumptions on Distribution Shifts**
>
> In SF-OW RFFI, distribution shifts stem primarily from dynamic wireless channels, while intrinsic hardware RFFs remain long-term invariant.
>
> Channel shifts impact classes asymmetrically: for unseen new classes, overall shifts bypass source-knowledge mismatch, leaving discovery unaffected. For old classes, training-deployment shifts are largely filtered out by our channel-invariant spectrogram preprocessing.
>
> To quantify sensitivity to shifts, we simulated channel fading via standard Rician models by varying K-factor from 50 (with strong Line-of-Sight (LoS) path) down to 0 (with only weak Non-Line-of-Sight (NLoS) paths).
>
> |K|50|20|10|5|2|1|0.5|0|
> |-|-|-|-|-|-|-|-|-|
> |Old|99.57|99.41|99.66|99.07|97.92|97.86|95.62|94.97|
> |New|99.81|99.49|99.75|99.94|99.81|99.56|99.49|99.81|
> |All|99.65|99.44|99.69|99.35|98.54|98.42|96.90|96.56|
>
> Results show that our framework is insensitive to moderate-to-strong channel shifts. Even under severe NLoS fading, overall accuracy remains >96%. Moreover, performance fluctuations are mostly confined to old classes, while new classes exhibit immunity, consistent with the asymmetric impact.
> >**Q3. Method Positioning**
>
> The confusion-forgetting dilemma in SF-OW is a universal ML challenge, merely amplified by high-similarity noisy RFFs. Conceptually, our framework provides a new perspective for general OWR: **shifting from passive parameter-space competition to proactive output-geometry design.**
>
> **1. From bottom-up self-clustering to top-down geometric guidance.**
> Existing OWR/GCD methods (e.g., ORCA, OpenLDN) rely heavily on similarity-driven data self-organization, making them prone to confusion or collapse. Our core insight is to proactively construct an optimal classifier-head geometry a priori, so as to provide robust separation guidance for unsupervised features from the output end.
>
> **2. From novel discovery only to source-free discovery-and-retention.**
> Existing OWR assumes source-data access (ignoring forgetting), while standard CL methods (e.g., GPM, EWC) still require source-data statistics for anti-forgetting. Thus, SF-OW can't be solved by trivially combining OWR and CL. **Our method shifts memory retention from source-dependent parameter regularization to source-free structural isolation.** This unifies novel discovery and old-class retention within a single geometric framework to elegantly resolve the stability-plasticity dilemma.
> >**Q4. Generalization**
>
> We carefully select our evaluated datasets (App.B.1), which are most representative and widely recognized public benchmarks in RFFI research community, covering diverse protocols, hardware qualities, and RF conditions: WiSig (high-bandwidth WiFi), Oracle (bit-similar devices), and LoRa (low-power IoT). Our consistent SOTA improvements across them prove strong deployment generalization.
>
> Moreover, we conducted a systematic stress test under different AWGN noise levels:
>
> |SNR(dB)|40|30|20|10|0|
> |-|-|-|-|-|-|
> |All|98.73|99.46|99.83|97.10|92.03|
> |AUROC|99.91|99.95|99.95|98.74|94.16|
>
> Our framework exhibits excellent resilience, maintaining >92% acc even under extreme 0dB SNR. This robustness stems from robust signal preprocessing and IO-ETF's maximal geometric separation.
> >**Q5. Practical Deployment & Complexity**
>
> We have systematically evaluated both training and inference complexity along with practical concerns. Due to space limits, we kindly direct the reviewer to our detailed discussion in our response to *Reviewer e42A (W1)*.
> ***
> We are grateful for your constructive suggestions. We hope that most concerns have been addressed and are happy to engage in further discussion.

---

> > ### Author Rebuttal · Reviewer_B73a · 2026-04-02
> >
> > Thank you for the rebuttal. While the additional explanations and experiments are helpful, I still find it difficult to clearly understand the overall pipeline and how the different components interact.
> >
> > Since my main concern was about clarity and accessibility, and this remains only partially addressed, I maintain my original score.

---

> > > ### Author Response · Authors · 2026-04-06
> > >
> > > Dear Reviewer B73a,
> > >
> > > We sincerely thank you for recognizing our solid experiments and explanations. To thoroughly clarify the overall execution flow, our method is a **two-stage sequential pipeline**: first construct and freeze a specific classifier head, then train the backbone to align its output features to this fixed head.
> > > * *Remark.* Unlike traditional classification methods that jointly optimize the backbone and classifier head, **we explicitly decouple the task into two stages. The first stage determines which classifier head is optimal for the SF-OW task; the second stage entirely freezes it and formally trains the backbone.** In this sense, the optimization focus of the latter phase is shifted to the feature space.
> > >
> > > **1. One-Time Classifier Head Initialization (IO-ETF)**
> > >
> > > **In this phase, we construct and freeze a classifier head skeleton suitable for SF-OW classification.**
> > >
> > > Intuitively, since the old classes already occupy fixed locations in the feature space (the hypersphere), we need to carefully allocate new territory for the upcoming new classes. By optimizing Eq.(4), we calculate the most spacious spots in the remaining space, placing the new class centers (anchors/fixed prototype) as far away from old classes, and as far apart from each other, as possible to reduce future confusion.
> > >
> > > **Once calculated, these positions are frozen, acting as fixed targets for the backbone during the subsequent training phase. In summary, IO-ETF is not a parallel module to TLGA, but the geometric skeleton on which TLGA operates.**
> > > * *Remark.* According to Neural Collapse, well-trained features extracted by the backbone and their classifier weights naturally align to vectors on a hypersphere. We pre-define the target vectors as anchors before the backbone starts learning.
> > >
> > > **2. Formal Training Phase (TLGA)**
> > >
> > > **With the classifier head frozen, this phase updates only the backbone.** The goal is for the backbone to stably align its output features to these predefined anchors. This process executes the implicit clustering logic: after alignment, each sample is classified to its nearest anchor (highest cosine similarity).
> > >
> > > **Within TLGA, three components interact via a two-layered dependency built on pseudo-labels**: Layer 1 generates pseudo-labels from backbone features (BEOT), and Layer 2 uses these pseudo-labels to compute two losses to optimize the backbone, forming a closed loop.
> > > * **Layer 1 (Foundation) BEOT Submodule**:
> > > It extracts features from the backbone to generate pseudo-labels for each sample. This submodule is thus responsible for determining *the target anchor for each feature from the set of predefined anchors*.
> > > * **Layer 2 (Execution) PMA & RAPR Submodules**:
> > > Both submodules are built on these pseudo-labels, applying different loss constraints to different targets (new/old classes) in parallel:
> > >   * **PMA (Pulling New Classes):** Based on the pseudo-labels, it maximizes the similarity between the feature and its target anchor while preserving local neighborhoods. This forms the alignment loss $L_{align}$, which drives progressive feature-to-anchor alignment for new classes.
> > >   * **RAPR (Locking Old Classes):** Using the pseudo-labels, it filters out high-confidence features around old anchors and locks them in place with retention loss $L_{ret}$. This prevents old-class representation drift due to PMA's pulling.
> > >
> > > Finally, the combined loss from the two submodules **backpropagates to update only the learnable backbone, while the IO-ETF anchors remain frozen throughout** (forming a positive loop: improved features yield more accurate pseudo-labels with the BEOT submodule). The interaction can be summarized as: **BEOT acts as the foundation to identify targets, while PMA and RAPR execute the movement: RAPR anchors old classes in their corresponding places, and PMA pulls new classes to their new anchors.**
> > >
> > > In essence, **classification for old and new classes is achieved through feature-space alignment**. Since phase 1 (IO-ETF) has already pre-allocated anchors for both old and new classes, the backbone's only task is to align features to these anchors. Once aligned, old devices recover their identities by reserving their places near their corresponding original anchors, whereas new devices are automatically detected and classified upon being assigned to newly reserved anchors.
> > >
> > > **Core Insight.** This duty separation introduces a new perspective for SF-OW: **top-down geometric guidance**. Instead of jointly training the full network with blind feature self-clustering, we predefine the desired optimal output, and then reversely guide the backbone to match it. This elegantly unifies the forgetting-confusion dilemma into classifier design, achieving a >7% boost in H-score.
> > > ***
> > > Thank you again for your valuable feedback, and we sincerely hope this detailed explanation could clarify the pipeline and is helpful in understanding. Please do not hesitate to let us know if any additional information is needed.

---

### Official Review · Reviewer_w7w6 · 2026-03-12

**Soundness:** 3
**Presentation:** 4
**Significance:** 3
**Originality:** 3
**Overall Recommendation:** 5
**Confidence:** 3

**Summary:**

This paper addresses the Source-Free Open-World (SF-OW) Radio Frequency Fingerprint Identification (RFFI) task. The authors identify a stability-plasticity dilemma where signal similarity causes confusion among new device classes, while the absence of original source data leading to catastrophic forgetting of old classes. To resolve this, they propose Incremental Orthogonal ETF (IO-ETF) that utilizes output-space geometry to induce parameter separation. They also introduce Triple-Level Geometric Alignment (TLGA), which uses semantic optimal transport, manifold progressive anchoring, and subspace retention to align unlabeled streams to the proposed geometric skeleton.

**Compliance With Llm Reviewing Policy:**

Affirmed.

**Final Justification:**

Most of my concerns of the paper are addressed, with the rest of them hard to be resolved during the rebuttal period. Thus I am raising my score.

**Key Questions For Authors:**

1. (p4, l207) The problem definition assumes the number of new classes (Knew) is known. This is a significant heuristic that simplifies the "Open-World" aspect. In true open-world scenarios, the number of novel categories is rarely known a priori. While the authors provide a sensitivity analysis in Appendix C.4, the core framework's reliance on a fixed-dimension geometric skeleton remains a practical limitation.
2. (p4, l200) The authors acknowledge that analytical Simplex ETFs require d>=K-1. While they propose an optimization-based construction to handle saturated regimes (K > d), the theoretical "worst-case angular margin" guarantee from Proposition 3.2 weakens significantly as the hypersphere becomes overcrowded.
3. The proposed solution involves a two-part solution (IO-ETF and TLGA) consisting of multiple optimization loops, iterations, and tri-level alignment. This adds substantial implementation and potentially computational complexity compared to standard self-training baselines.
4. (p5, l227, eq5) The authors mention that ɛ controls smoothness in the Balanced Entropic Optimal Transport (BEOT). However, there is no quantitative discussion or ablation on how the choice of ɛ specifically impacts the stability of pseudo-labeling in highly noisy wireless environments.
5. (p8, l432) The authors note an approximate 5% accuracy drop when removing the RAPR component. Given that RAPR relies on a confidence threshold (γ = 0.6), the review finds the justification for this specific threshold largely heuristic. A more rigorous analysis of how the threshold selection affects the self-drift of old features is missing.
6. The implementation details for Sinkhorn regularization, batch sizes for OT, memory bank size/update, thresholds κ, γ for RAPR are described in the appendix. There should be hyperparameter table in the main text for reproducibility.
7. There are minor typos here and there, suggests that the author might have rushed and did not proofread the manuscript properly. (“Collaspe” in Fig. 2, level 1). The heavy notations should be adjusted for better readability.

**Limitations:**

yes

**Strengths And Weaknesses:**

## Pros
1. The paper identifies that standard closed-set RFFI is insufficient for real-world deployments involving a continuous influx of unregistered devices and strict privacy constraints.
2. The shift from parameter-space competition to output-space geometry is well-motivated by the phenomenon of Neural Collapse.
3. The IO-ETF framework's use of inter-block orthogonality for gradient isolation is an intuitive way to mitigate interference without historical data replay.
4. The method achieves state-of-the-art results across three benchmarks (WiSig, Oracle, LoRa).
5. The paper is generally clear and well-organized, and visualizations help convey the core idea effectively.
\end{enumerate}

## Cons
Please refer to the questions.

---

> ### Author Rebuttal · Authors · 2026-03-31
>
> Dear Reviewer w7w6,
>
> Thank you for your insightful review and suggestions, and we are grateful for your recognition of our contributions. We respond to your concerns below.
> >**Q1. Assumption of Known $K_{new}$**
>
> While assuming a known $K_{new}$ has been a common and standard simplification in OW and GCD literature to isolate variables and focus on core representation-alignment challenges, we fully agree that relaxing this assumption is important for practical deployment.
>
> Therefore, we wish to clarify that our framework is not overly sensitive to $K_{new}$ errors. It only requires a coarse but not excessively low estimate to maintain competitive performance (App.C.4). In practice, the total class count $\hat{K}$ can be inferred using lightweight unsupervised clustering estimators (e.g., DPMM and K-means with silhouette-based K selection), deriving $\hat{K}\_{new}=\hat{K}-K\_{old}$ prior to skeleton instantiation.
>
> |Dataset(K)|Estimator|$\hat{K}$|Acc(%)|AUROC(%)|
> |-|-|-|-|-|
> |WiSig(6)|DPMM|6|99.96|99.73|
> ||K-means|6|99.96|99.73|
> |Oracle(15)|DPMM|19|90.37|94.59|
> ||K-means|14|86.52|94.49|
> |LoRa(25)|DPMM|23|91.71|91.97|
> ||K-means|25|94.64|93.38|
>
> Results demonstrate that with the preliminary estimates, our scheme retains competitive performance.
>
> Notably, our geometric skeleton exhibits superior tolerance for over-estimation. A practical deployment strategy is to leave a moderate margin during estimation, allowing the skeleton to pre-allocate redundant anchors for potential new classes without disrupting the existing feature space, ultimately ensuring robust recognition.
> >**Q2. Adaptive Geometric Packing in Saturated Regimes**
>
> We agree that worst-case angular margin inevitably weakens as the hypersphere becomes crowded. Under Generalized Neural Collapse theory, this reflects a fundamental geometric bottleneck shared by all classifiers, since last-layer class representatives converge to equal norms and thus inherently reside on a bounded hypersphere.
>
> The advantage of IO-ETF is that it adaptively optimizes the packing to maximally mitigate this possible crowding. In the unsaturated regime, it recovers Simplex-ETF solution. Upon entering the saturated regime, the objective naturally shifts from simplex-style optimality to seeking the best feasible GNC low-coherence packing on the hypersphere via Eq.(4). This mechanism drives the automatic simplex-to-orthoplex transition (Fig.3b and App.C.1). We also validate its sustained performance advantages over alternative classifier construction strategies in highly crowded scenarios (Table 2 in the main text).
>
> Finally, we appreciate the reminder of extreme large-scale open-world scenarios (where K far exceeds the current feature dimension d=512). In such cases, increasing d serves as a natural knob to effectively alleviate spherical packing limits.
> >**Q3. Implementation and Computational Complexity**
>
> We thank the reviewer for raising this important concern. Due to strict character limits, we kindly direct the reviewer to our response to *Reviewer e42A (W1)* for a detailed quantitative analysis demonstrating our efficient deployment and manageable adaptation costs.
> >**Q4&5. Quantitative Discussion on ɛ and γ**
>
> * ɛ of BEOT
>
> We quantify the effect of ɛ and introduce the Pseudo-Label Flip Rate (PLFR) between adjacent epochs as a micro-stability metric.
>
> |ɛ|0.01|0.02|0.05|0.1|0.2|0.5|
> |-|-|-|-|-|-|-|
> |Old|92.24|96.93|99.98|99.32|99.66|99.78|
> |New|99.75|99.94|99.90|99.43|99.81|99.56|
> |PLFR|2.40|0.08|0.02|0.04|0.00|0.00|
>
> Small ɛ degrades OT assignment into noise-sensitive rigid matching. This spikes PLFR to 2.4% (severe label oscillation), which propagates noisy gradients and harms old-class retention. In contrast, a moderate ɛ (0.05) buffers noise, yielding smoother batch-wise matching, suppressing PLFR near 0 while robustly maintaining strong overall performance.
> * γ of RAPR
>
> To quantify how γ affects old-feature self-drift, we introduce the Feature Retention Score (FRS): the mean cosine alignment of identical old-class samples before/after training.
>
> |γ|0.3|0.4|0.5|0.6|0.7|0.8|0.9|
> |-|-|-|-|-|-|-|-|
> |Old|98.51|97.95|98.11|99.98|97.77|97.92|94.79|
> |New|99.94|99.75|99.81|99.90|99.75|99.75|99.81|
> |FRS|0.9060|0.9458|0.9413|0.9618|0.9451|0.9466|0.9401|
>
> Results show γ=0.6 optimizes purity-coverage trade-off, minimizing old-class self-drift. Lower thresholds over-include pseudo-old samples, polluting retention (FRS=0.906). Higher thresholds ensure purity but retain insufficient old-core anchors, weakening subspace preservation, dropping old acc (94.79% at γ=0.9).
> >**Q6&7. Paper Presentation and Readability**
>
> We will organize the hyperparameter table from appendix to the main text. Moreover, we will thoroughly proofread the manuscript and streamline notations to enhance readability.
> ***
> We sincerely thank the reviewer for your professional and insightful suggestions, and helping improve our work. We will make sure to add more discussions of these points in the revision.

---

> > ### Author Rebuttal · Reviewer_w7w6 · 2026-04-02
> >
> > I would like to thank the authors for the detailed response. Most of our concerns are addressed, so I am raising my score. I would still like to see how performance degrades as K significantly exceeds d, but exploring into this is probably hard for the time given.

---

> > > ### Author Response · Authors · 2026-04-06
> > >
> > > Dear Reviewer w7w6,
> > >
> > > We sincerely thank you for your strong support of our work and for your insightful curiosity regarding the K $\gg$ d regime. To explore this interesting question, we conducted an additional extreme over-saturation stress test.
> > >
> > > Under challenging fixed-class settings on Oracle (K=15), we aggressively reduce the feature dimension from d=12 down to d=6.
> > >
> > > * K=15 with 4 old and 11 new classes
> > >
> > > |Metric|d=6|d=8|d=10|d=12|
> > > |---|---|---|---|---|
> > > |Old|80.72|95.67|97.28|99.36|
> > > |New|61.92|61.07|63.58|67.99|
> > > |AUROC|75.04|79.84|81.05|85.79|
> > >
> > > * K=15 with 5 old and 10 new classes
> > >
> > > |Metric|d=6|d=8|d=10|d=12|
> > > |---|---|---|---|---|
> > > |Old|75.80|85.76|94.84|96.64|
> > > |New|67.14|69.84|74.11|79.71|
> > > |AUROC|68.54|79.53|88.03|90.39|
> > >
> > >
> > > As you astutely anticipated, performance inevitably drops as the hypersphere becomes extremely crowded, which is consistent with our explanation of the inherent bottlenecks of spherical packing. More importantly, even under the extreme d=6 setting, the framework remains far above random guessing, indicating that it still actively pushes new classes apart to maintain stable boundaries and seeks a feasible low-crowding arrangement.
> > >
> > > This also highlights that d is the key factor governing hypersphere capacity. Under Neural Collapse theory, features collapse to their centers with small intra-class variance, bounded within a spherical cap, which induces a required minimum angular separation $\theta$ between class centers. A standard spherical-cap packing argument then implies that the capacity to accommodate K classes grows rapidly with d: $K \le \left( \frac{1}{\sin(\theta/2)} \right)^{d-1}$.
> > >
> > > Therefore, the degradation observed is actually a harsh low-dimensional proxy: with d=512 as in our paper, reaching a comparable degree of crowding would typically require a vastly larger K. We are also collecting a larger-scale RFF dataset to contribute to the community for more exploration in the future.
> > >
> > > We truly appreciate your professional insights and deep interest in our work!

---

### Decision · Program_Chairs · 2026-04-30

**Decision:**

Accept (regular)

**Comment:**

This paper considers the problem of Source-Free Open-World Radio Frequency Fingerprint Identification, where a model trained on known devices must adapt to unlabeled stream from known and unknown devices. The authors propose a neural collapse inspired framework and demonstrate its effectiveness in identification experimentally. All the reviews are positive about the contributions of this paper and therefore I recommend acceptance. That said, I encourage the authors to revisit their paper in light of the reviewers' comments, especially regarding accessibility and readability for a broader audience, and the concerns regarding the overhead of the framework and its failure cases.